# ON COMPUTATION AND GENERALIZATION OF GENERATIVE ADVERSARIAL IMITATION LEARNING

**Minshuo Chen**[*]  **Yizhou Wang**[†]  **Tianyi Liu**[*]  **Zhuoran Yang**[‡]  **Xingguo Li**[‡]
**Zhaoran Wang**[**]  **Tuo Zhao**[*]
[*] Georgia Tech, [†] Xian Jiaotong University, [‡] Princeton University, [**] Northwestern University

## ABSTRACT

Generative Adversarial Imitation Learning (GAIL) is a powerful and practical approach for learning sequential decision-making policies. Different from Reinforcement Learning (RL), GAIL takes advantage of demonstration data by experts (e.g., human), and learns both the policy and reward function of the unknown environment. Despite the significant empirical progresses, the theory behind GAIL is still largely unknown. The major difficulty comes from the underlying temporal dependency of the demonstration data and the minimax computational formulation of GAIL without convex-concave structure. To bridge such a gap between theory and practice, this paper investigates the theoretical properties of GAIL. Specifically, we show: (1) For GAIL with general reward parameterization, the generalization can be guaranteed as long as the class of the reward functions is properly controlled; (2) For GAIL, where the reward is parameterized as a reproducing kernel function, GAIL can be efficiently solved by stochastic first order optimization algorithms, which attain sublinear convergence to a stationary solution. To the best of our knowledge, these are the first results on statistical and computational guarantees of imitation learning with reward/policy function approximation. Numerical experiments are provided to support our analysis.

## 1 INTRODUCTION

As various robots (Tail et al., 2018), self-driving cars (Kuefler et al., 2017), unmanned aerial vehicles (Pfeiffer et al., 2018) and other intelligent agents are applied to complex and unstructured environments, programming their behaviors/policy has become increasingly challenging. These intelligent agents need to accommodate a huge number of tasks with unique environmental demands. To address these challenges, many reinforcement learning (RL) methods have been proposed for learning sequential decision-making policies (Sutton et al., 1998; Kaelbling et al., 1996; Mnih et al., 2015). These RL methods, however, heavily rely on human expert domain knowledge to design proper reward functions. For complex tasks, which are often difficult to describe formally, these RL methods become impractical.

The Imitation Learning (IL, Argall et al. (2009); Abbeel & Ng (2004)) approach is a powerful and practical alternative to RL. Rather than having a human expert handcrafting a reward function for learning the desired policy, the imitation learning approach only requires the human expert to demonstrate the desired policy, and then the intelligent agent (a.k.a. learner) learns to match the demonstration. Most of existing imitation learning methods fall in the following two categories:

• Behavioral Cloning (BC, Pomerleau (1991)). BC treats the IL problem as supervised learning. Specifically, it learns a policy by fitting a regression model over expert demonstrations, which directly maps states to actions. Unfortunately, BC has a fundamental drawback. Recall that in supervised learning, the distribution of the training data is decoupled from the learned model, whereas in imitation learning, the agent's policy affects what state is queried next. The mismatch between training and testing distributions, also known as covariate shift (Ross & Bagnell, 2010; Ross et al., 2011), yields significant compounding errors. Therefore, BC often suffers from poor generalization.

• Inverse Reinforcement Learning (IRL, Russell (1998); Ng et al. (2000); Finn et al. (2016); Levine & Koltun (2012)). IRL treats the IL problem as bi-level optimization. Specifically, it finds a reward

function, under which the expert policy is uniquely optimal. Though IRL does not have the error compounding issue, its computation is very inefficient. Many existing IRL methods need to solve a sequence of computationally expensive reinforcement learning problems, due to their bi-level optimization nature. Therefore, they often fail to scale to large and high dimensional environments.

More recently, Ho & Ermon (2016) propose a Generative Adversarial Imitation Learning (GAIL) method, which obtains significant performance gains over existing IL methods in imitating complex expert policies in large and high-dimensional environments. GAIL generalizes IRL by formulating the IL problem as minimax optimization, which can be solved by alternating gradient-type algorithms in a more scalable and efficient manner.

Specifically, we consider an infinite horizon Markov Decision Process (MDP), where $\mathcal{S}$ denotes the state space, $\mathcal{A}$ denotes the action space, $P$ denotes the Markov transition kernel, $r^*$ denotes the reward function, and $p_0$ denotes the distribution of the initial state. We assume that the Markov transition kernel $P$ is fixed and there is an unknown expert policy $\pi^* \colon \mathcal{S} \to \mathcal{P}(\mathcal{A})$, where $\mathcal{P}(\mathcal{A})$ denotes the set of distributions over the action space. As can be seen, $\{s_t\}_{t=0}^{T-1}$ essentially forms a Markov chain with the transition kernel induced by $\pi^*$ as $P^{\pi^*}(s, s') = \sum_{a \in \mathcal{A}} \pi^*(a \,|\, s) \cdot P(s' \,|\, s, a)$. Given $n$ demonstration trajectories from $\pi^*$ denoted by $\{s_t^{(i)}, a_t^{(i)}\}_{t=0}^{T-1}$, where $i = 1, ..., n$, $s_0 \sim p_0$, $a_t \sim \pi^*(\cdot \,|\, s_t)$, and $s_{t+1} \sim P(\cdot \,|\, s_t, a_t)$, GAIL aims to learn $\pi^*$ by solving the following minimax optimization problem,

$$\min_\pi \max_{r \in \mathcal{R}} \left[ \mathbb{E}_\pi r(s, a) - \mathbb{E}_{\pi_n^*} r(s, a) \right], \tag{1}$$

where $\mathbb{E}_\pi[r(s, a)] = \lim_{T \to \infty} \mathbb{E}[\frac{1}{T} \sum_{t=0}^{T-1} r(s_t, a_t) | \pi]$ denotes the average reward under the policy $\pi$ when the reward function is $r$, and $\mathbb{E}_{\pi_n^*}[r(s, a)] = \frac{1}{nT} \sum_{i=1}^{n} \sum_{t=0}^{T-1} [r(s_t^{(i)}, a_t^{(i)})]$ denotes the empirical average reward over the demonstration trajectories. As shown in (1), GAIL aims to find a policy, which attains an average reward similar to that of the expert policy with respect to any reward belonging to the function class $\mathcal{R}$.

For large and high-dimensional imitation learning problems, we often encounter infinitely many states. To ease computation, we need to consider function approximations. Specifically, suppose that for every $s \in \mathcal{S}$ and $a \in \mathcal{A}$, there are feature vectors $\psi_s \in \mathbb{R}^{d_\mathcal{S}}$ and $\psi_a \in \mathbb{R}^{d_\mathcal{A}}$ associated with $a$ and $s$, respectively. Then we can approximate the policy and reward as

$$\pi(\cdot | s) = \widetilde{\pi}_\omega(\psi_s) \quad \text{and} \quad r(s, a) = \widetilde{r}_\theta(\psi_s, \psi_a),$$

where $\widetilde{\pi}$ and $\widetilde{r}$ belong to certain function classes (e.g. reproducing kernel Hilbert space or deep neural networks, Ormoneit & Sen (2002); LeCun et al. (2015)) associated with parameters $\omega$ and $\theta$, respectively. Accordingly, we can optimize (1) with respect to the parameters $\omega$ and $\theta$ by scalable alternating gradient-type algorithms.

Although GAIL has achieved significant empirical progresses, its theoretical properties are still largely unknown. There are three major difficulties when analyzing GAIL: 1). There exists temporal dependency in the demonstration trajectories/data due to their sequential nature (Howard, 1960; Puterman, 2014; Abounadi et al., 2001); 2). GAIL is formulated as a minimax optimization problem. Most of existing learning theories, however, focus on empirical risk minimization problems, and therefore are not readily applicable (Vapnik, 2013; Mohri et al., 2018; Anthony & Bartlett, 2009); 3). The minimax optimization problem in (1) does not have a convex-concave structure, and therefore existing theories in convex optimization literature cannot be applied for analyzing the alternating stochastic gradient-type algorithms (Willem, 1997; Ben-Tal & Nemirovski, 1998; Murray & Overton, 1980; Chambolle & Pock, 2011; Chen et al., 2014). Some recent results suggest to use stage-wise stochastic gradient-type algorithms (Rafique et al., 2018; Dai et al., 2017). More specifically, at every iteration, they need to solve the inner maximization problem up to a high precision, and then apply stochastic gradient update to the outer minimization problem. Such algorithms, however, are rarely used by practitioners, as they are inefficient in practice (due to the computationally intensive inner maximization).

To bridge such a gap between practice and theory, we establish the generalization properties of GAIL and the convergence properties of the alternating mini-batch stochastic gradient algorithm for solving (1). Specifically, our contributions can be summarized as follows:

• We formally define the generalization of GAIL under the "so-called" $\mathcal{R}$-reward distance, and then show that the generalization of GAIL can be guaranteed under reward distance as long as the class of the reward functions is properly controlled;

• We provide sufficient conditions, under which an alternating mini-batch stochastic gradient algorithm can efficiently solve the minimax optimization in (1), and attains sublinear convergence to a stationary solution.

To the best of our knowledge, these are the first results on statistical and computational theories of imitation learning with reward/policy function approximations.

Our work is related to Syed et al. (2008); Cai et al. (2019). Syed et al. (2008) study the generalization and computational properties of apprenticeship learning. Since they assume that the state space of the underlying Markov decision process is finite, they do not consider any reward/policy function approximations; Cai et al. (2019) study the computational properties of imitation learning under a simple control setting. Their assumption on linear policy and quadratic reward is very restrictive, and does not hold for many real applications.

**Notation.** Given a vector $x = (x_1, ..., x_d)^\top \in \mathbb{R}^d$, we define $\|x\|_2^2 = \sum_{j=1}^{d} x_j^2$. Given a function $f : \mathbb{R}^d \mapsto \mathbb{R}$, we denote its $\ell_\infty$ norm as $\|f\|_\infty = \max_x |f(x)|$.

## 2 GENERALIZATION OF GAIL

To analyze the generalization properties of GAIL, we first assume that we can access an infinite number of the expert's demonstration trajectories (underlying population), and that the reward function is chosen optimally within some large class of functions. This allows us to remove the maximum operation from (1), which leads to an interpretation of how and in what sense the resulting policy is close to the true expert policy. Before we proceed, we first introduce some preliminaries.

**Definition 1** (Stationary Distribution). Note that any policy $\pi$ induces a Markov chain on $\mathcal{S} \times \mathcal{A}$. The transition kernel is given by

$$P_\pi(s', a' \,|\, s, a) = \pi(a' \,|\, s') \cdot P(s' \,|\, s, a), \quad \forall (s, a), (s', a') \in \mathcal{S} \times \mathcal{A}.$$

When such a Markov chain is aperiodic and recurrent, we denote its stationary distribution as $\rho_\pi$.

Note that a policy $\pi$ is uniquely determined by its stationary distribution $\rho_\pi$ in the sense that

$$\pi(a \,|\, s) = \rho_\pi(s, a) / \sum_{a \in \mathcal{A}} \rho_\pi(s, a).$$

Then we can write the expected average reward of $r(s, a)$ under the policy $\pi$ as

$$\mathbb{E}_\pi[r(s, a)] = \lim_{T \to \infty} \mathbb{E}\big[\tfrac{1}{T} \sum_{t=0}^{T-1} r(s_t, a_t) \big| \pi\big] = \mathbb{E}_{\rho_\pi}\big[r(s, a)\big] = \sum_{(s,a) \in \mathcal{S} \times \mathcal{A}} \rho_\pi(s, a) \cdot r(s, a).$$

We further define the $\mathcal{R}$-distance between two policies $\pi$ and $\pi'$ as follows.

**Definition 2.** Let $\mathcal{R}$ denote a class of symmetric reward functions from $\mathcal{S} \times \mathcal{A}$ to $\mathbb{R}$, i.e., if $r \in \mathcal{R}$, then $-r \in \mathcal{R}$. Given two policy $\pi'$ and $\pi$, the $\mathcal{R}$-distance for GAIL is defined as

$$d_\mathcal{R}(\pi, \pi') = \sup_{r \in \mathcal{R}} [\mathbb{E}_\pi r(s, a) - \mathbb{E}_{\pi'} r(s, a)].$$

The $\mathcal{R}$-distance over policies for Markov decision processes is essentially an Integral Probability Metric (IPM) over stationary distributions (Müller, 1997). For different choices of $\mathcal{R}$, we have various $\mathcal{R}$-distances. For example, we can choose $\mathcal{R}$ as the class of all 1-Lipschitz continuous functions, which yields that $d_\mathcal{R}(\pi, \pi')$ is the Wasserstein distance between $\rho_\pi$ and $\rho_{\pi'}$ (Vallender, 1974). For computational convenience, GAIL and its variants usually choose $\mathcal{R}$ as a class of functions from some reproducing kernel Hilbert space, or a class of neural network functions.

**Definition 3.** Given $n$ demonstration trajectories from time 0 to $T-1$ obtained by an expert policy $\pi^*$ denoted by $(s_t^{(i)}, a_t^{(i)})_{t=0}^{T-1}$, where $i = 1, ..., n$, a policy $\widehat{\pi}$ learned by GAIL generalizes under the $\mathcal{R}$-distance $d_\mathcal{R}(\cdot, \cdot)$ with generalization error $\epsilon$, if with high probability, we have

$$|d_\mathcal{R}(\pi_n^*, \widehat{\pi}) - d_\mathcal{R}(\pi^*, \widehat{\pi})| \leq \epsilon,$$

where $d_\mathcal{R}(\pi_n^*, \widehat{\pi})$ is the empirical $\mathcal{R}$-distance between $\pi^*$ and $\widehat{\pi}$ defined as

$$d_\mathcal{R}(\pi_n^*, \widehat{\pi}) = \sup_{r \in \mathcal{R}} [\mathbb{E}_{\pi_n^*} r(s, a) - \mathbb{E}_{\widehat{\pi}} r(s, a)] \quad \text{with} \quad \mathbb{E}_{\pi_n^*}[r(s, a)] = \tfrac{1}{nT} \sum_{i=1}^{n} \sum_{t=0}^{T-1} [r(s_t^{(i)}, a_t^{(i)})].$$

The generalization of GAIL implies that the $\mathcal{R}$-distance between the expert policy $\pi^*$ and the learned policy $\widehat{\pi}$ is close to the empirical $\mathcal{R}$-distance between them. Our analysis aims to prove the former distance to be small, whereas the latter one is what we attempts to minimize in practice.

We then introduce the assumptions on the underlying Markov decision process and expert policy.

**Assumption 1.** Under the expert policy $\pi^*$, $(s_t, a_t)_{t=0}^{T-1}$ forms a stationary and exponentially $\beta$-mixing Markov chain, i.e.,

$$\beta(k) = \sup_n \mathbb{E}_{B \in \sigma_0^n} \sup_{A \in \sigma_{n+k}^\infty} |\mathbb{P}(A|B) - \mathbb{P}(A)| \leq \beta_0 \exp(-\beta_1 k^\alpha),$$

where $\beta_0, \beta_1, \alpha$ are positive constants, and $\sigma_i^j$ is the $\sigma$-algebra generated by $(s_t, a_t)_{t=i}^j$ for $i \leq j$.

Moreover, for every $s \in \mathcal{S}$ and $a \in \mathcal{A}$, there are feature vectors $\psi_s \in \mathbb{R}^{d_{\mathcal{S}}}$ and $\psi_a \in \mathbb{R}^{d_{\mathcal{A}}}$ associated with $a$ and $s$, respectively, and $\psi_s$ and $\psi_a$ are uniformly bounded, where

$$\|\psi_s\|_2 \leq 1 \quad \text{and} \quad \|\psi_a\|_2 \leq 1, \quad \forall s \in \mathcal{S} \quad \text{and} \quad \forall a \in \mathcal{A}.$$

Assumption 1 requires the underlying MDP to be ergodic (Levin & Peres, 2017), which is a commonly studied assumption in exiting reinforcement learning literature on maximizing the expected average reward (Strehl & Littman, 2005; Li et al., 2011; Brafman & Tennenholtz, 2002; Kearns & Singh, 2002). The feature vectors associated with $a$ and $s$ allow us to apply function approximations to parameterize the reward and policy functions. Accordingly, we write the reward function as $r(s, a) = \widetilde{r}(\psi_s, \psi_a)$, which is assumed to be bounded.

**Assumption 2.** The reward function class is uniformly bounded, i.e., $\|r\|_\infty \leq B_r$ for any $r \in \mathcal{R}$.

Now we proceed with our main result on generalization properties of GAIL. We use $\mathcal{N}(\mathcal{R}, \epsilon, \|\cdot\|_\infty)$ to denote the covering number of the function class $\mathcal{R}$ under the $\ell_\infty$ distance $\|\cdot\|_\infty$.

**Theorem 1** (Main Result). Suppose Assumptions 1-2 hold, and the policy learned by GAIL satisfies

$$d_{\mathcal{R}}(\pi_n^*, \widehat{\pi}) - \inf_\pi d_{\mathcal{R}}(\pi_n^*, \pi) < \epsilon,$$

where the infimum is taken over all possible learned policies. Then with probability at least $1 - \delta$ over the joint distribution of $\{(a_t^{(i)}, s_t^{(i)})_{t=0}^{T-1}\}_{i=1}^n$, we have

$$d_{\mathcal{R}}(\pi^*, \widehat{\pi}) - \inf_\pi d_{\mathcal{R}}(\pi^*, \pi) \leq O\left(\frac{B_r}{\sqrt{nT/\zeta}}\sqrt{\log \mathcal{N}\left(\mathcal{R}, \sqrt{\frac{\zeta}{nT}}, \|\cdot\|_\infty\right)} + B_r\sqrt{\frac{\log(1/\delta)}{nT/\zeta}}\right) + \epsilon,$$

where $\zeta = (\beta_1^{-1} \log \frac{\beta_0 T}{\delta})^{\frac{1}{\alpha}}$.

Theorem 1 implies that the policy $\widehat{\pi}$ learned by GAIL generalizes as long as the complexity of the function class $\mathcal{R}$ is well controlled. To the best of our knowledge, this is the first result on the generalization of imitation learning with function approximations. As the proof of Theorem 1 is involved, we only present a sketch due to space limit. More details are provided in Appendix A.1.

**Proof Sketch.** Our analysis relies on characterizing the concentration property of the empirical average reward under the expert policy. For notational simplicity, we define

$$\phi = \mathbb{E}_{\pi^*} r(s, a) - \frac{1}{nT} \sum_{i=1}^n \sum_{t=0}^{T-1} r(s_t^{(i)}, a_t^{(i)}).$$

The key challenge comes from the fact that $(s_t^{(i)}, a_t^{(i)})$'s are dependent. To handle such a dependency, we adopt the independent block technique from Yu (1994). Specifically, we partition every trajectory into disjoint blocks (where the block size is of the order $O((\log(T) + \log(1/\delta))^{1/\alpha})$, and construct two separable trajectories: One contains all blocks with odd indices (denoted by $\mathcal{B}_{\text{odd}}$), and the other contains all those with even indices (denoted by $\mathcal{B}_{\text{even}}$). We define

$$\phi_1 = \mathbb{E}_{\pi^*} r(s, a) - \frac{2}{nT} \sum_{i=1}^n \sum_{(s_t^{(i)}, a_t^{(i)}) \in \mathcal{B}_{\text{odd}}} r(s_t^{(i)}, a_t^{(i)}),$$

and analogously for $\phi_2$ with $(s_t^{(i)}, a_t^{(i)}) \in \mathcal{B}_{\text{even}}$. Then we have

$$\mathbb{P}(\sup_{r \in \mathcal{R}} \phi \geq \varepsilon) \leq \mathbb{P}(\sup_{r \in \mathcal{R}} \tfrac{\phi_1}{2} + \sup_{r \in \mathcal{R}} \tfrac{\phi_2}{2} \geq \varepsilon) \leq \mathbb{P}(\sup_{r \in \mathcal{R}} \phi_1 \geq \varepsilon) + \mathbb{P}(\sup_{r \in \mathcal{R}} \phi_2 \geq \varepsilon).$$

We consider a block-wise independent counterpart of $\phi_1$ denoted by $\widetilde{\phi}_1$, where each block is sampled independently from the same Markov chain as $\phi_1$, i.e., $\widetilde{\phi}_1$ has independent blocks of samples from the same exponentially $\beta$-mixing Markov chain . Accordingly, we denote $\widetilde{\phi}_1$ as

$$\widetilde{\phi}_1 = \mathbb{E}_{\pi^*} r(s,a) - \frac{2}{nT} \sum_{i=1}^n \sum_{(s_t^{(i)}, a_t^{(i)}) \in \widetilde{\mathcal{B}}_{\text{odd}}} r(s_t^{(i)}, a_t^{(i)}),$$

where $\widetilde{\mathcal{B}}_{\text{odd}}$ denotes i.i.d. blocks of samples. Now we bound the difference between $\phi_1$ and $\widetilde{\phi}_1$ by

$$\mathbb{P}(\sup_{r \in \mathcal{R}} \phi_1 - \mathbb{E}[\sup_{r \in \mathcal{R}} \widetilde{\phi}_1] \geq \varepsilon - \mathbb{E}[\sup_{r \in \mathcal{R}} \widetilde{\phi}_1]) \leq \mathbb{P}(\sup_{r \in \mathcal{R}} \widetilde{\phi}_1 - \mathbb{E}[\sup_{r \in \mathcal{R}} \widetilde{\phi}_1] \geq \varepsilon - \mathbb{E}[\sup_{r \in \mathcal{R}} \widetilde{\phi}_1])$$
$$+ C\beta T/(\log(T) + \log(1/\delta))^{1/\alpha},$$

where $C$ is a constant, and $\beta$ is the mixing coefficient, and $\mathbb{P}(\sup_{r \in \mathcal{R}} \widetilde{\phi}_1 - \mathbb{E}[\sup_{r \in \mathcal{R}} \widetilde{\phi}_1] \geq \varepsilon - \mathbb{E}[\sup_{r \in \mathcal{R}} \widetilde{\phi}_1])$ can be bounded using the empirical process technique for independent random variables. The details of the above inequality can be found in Corollary 3 in Appendix A.1, where the proof technique is adapted from Lemma 1 in Mohri & Rostamizadeh (2009). Let $\widetilde{\phi}_2$ be defined analogously as $\widetilde{\phi}_1$. With a similar argument further applied to $\phi_2$ and $\widetilde{\phi}_2$, we obtain

$$\mathbb{P}(\sup_{r \in \mathcal{R}} \phi \geq \varepsilon) \leq 2\mathbb{P}(\sup_{r \in \mathcal{R}} \widetilde{\phi}_1 - \mathbb{E}[\sup_{r \in \mathcal{R}} \widetilde{\phi}_1] \geq \varepsilon - \mathbb{E}[\sup_{r \in \mathcal{R}} \widetilde{\phi}_1]) + 2C\beta T/(\log(T) + \log(1/\delta))^{1/\alpha}.$$

The rest of our analysis follows the PAC-learning framework using Rademacher complexity and is omitted (Mohri et al., 2018). We complete the proof sketch. $\qquad\square$

**Example 1: Reproducing Kernel Reward Function**. One popular option to parameterize the reward by functions is the reproducing kernel Hilbert space (RKHS, Kim & Park (2018); Li et al. (2018)). There have been several implementations of RKHS, and we consider the feature mapping approach. Specifically, we consider $g : \mathbb{R}^{d_\mathcal{S}} \times \mathbb{R}^{d_\mathcal{A}} \to \mathbb{R}^q$, and the reward can be written as

$$r(s,a) = \widetilde{r}_\theta(\psi_s, \psi_a) = \theta^\top g(\psi_s, \psi_a),$$

where $\theta \in \mathbb{R}^q$. We require $g$ to be Lipschitz continuous with respect to $(\psi_a, \psi_s)$.

**Assumption 3.** The feature mapping $g$ satisfies $g(0,0) = 0$, and there exists a constant $\rho_g$ such that for any $\psi_a, \psi_a', \psi_s$ and $\psi_s'$, we have

$$\|g(\psi_s, \psi_a) - g(\psi_s', \psi_a')\|_2^2 \leq \rho_g \sqrt{\|\psi_s - \psi_s'\|_2^2 + \|\psi_a - \psi_a'\|_2^2}.$$

Assumption 3 is mild and satisfied by popular feature mappings, e.g., random Fourier feature mapping[1] (Rahimi & Recht, 2008; Bach, 2017). The next corollary presents the generalization bound of GAIL using feature mapping.

**Corollary 1.** Suppose $\|\theta\|_2 \leq B_\theta$. For large enough $n$ and $T$, with probability at least $1 - \delta$ over the joint distribution of $\{(a_t^{(i)}, s_t^{(i)})_{t=0}^{T-1}\}_{i=1}^n$, we have

$$d_\mathcal{R}(\pi^*, \widehat{\pi}) - \inf_\pi d_\mathcal{R}(\pi^*, \pi) \leq O\left(\frac{\rho_g B_\theta}{\sqrt{nT/\zeta}} \sqrt{q \log\left(\rho_g B_\theta \sqrt{nT/\zeta}\right)} + \rho_g B_\theta \sqrt{\frac{\log(1/\delta)}{nT/\zeta}}\right) + \epsilon.$$

Corollary 1 indicates that with respect to a class of properly normalized reproducing kernel reward functions, GAIL generalizes in terms of the $\mathcal{R}$-distance.

**Example 2: Neural Network Reward Function**. Another popular option to parameterize the reward function is to use neural networks. Specifically, let $\sigma(v) = [\max\{v_1, 0\}, ..., \max\{v_d, 0\}]^\top$ denote the ReLU activation for $v \in \mathbb{R}^d$. We consider a $D$-layer feedforward neural network with ReLU activation as follows,

$$r(s,a) = \widetilde{r}_\mathcal{W}(\psi_s, \psi_a) = W_D^\top \sigma(W_{D-1}\sigma(...\sigma(W_1[\psi_a^\top, \psi_s^\top]^\top))),$$

where $\mathcal{W} = \{W_k \mid W_k \in \mathbb{R}^{d_{k-1} \times d_k}, \ k = 1, ..., D-1, \ W_D \in \mathbb{R}^{d_{D-1}}\}$ and $d_0 = d_\mathcal{A} + d_\mathcal{S}$. The next corollary presents the generalization bound of GAIL using neural networks.

---

[1] More precisely, Assumption 3 actually holds with overwhelming probability over the distribution of the random mapping.

**Corollary 2.** Suppose $\|W_i\|_2 \leq 1$, where $i = 1, ..., D$. For large enough $n$ and $T$, with probability at least $1 - \delta$ over the joint distribution of $\{(a_t^{(i)}, s_t^{(i)})_{t=0}^{T-1}\}_{i=1}^n$, we have

$$d_{\mathcal{R}}(\pi^*, \widehat{\pi}) - \inf_{\pi} d_{\mathcal{R}}(\pi^*, \pi) \leq O\left(\frac{1}{\sqrt{nT/\zeta}}\sqrt{d^2 D \log\left(D\sqrt{dnT/\zeta}\right)} + \sqrt{\frac{\log(1/\delta)}{nT/\zeta}}\right) + \epsilon.$$

Corollary 2 indicates that with respect to a class of properly normalized neural network reward functions, GAIL generalizes in terms of the $\mathcal{R}$-distance.

**Remark 1** (**The Tradeoff between Generalization and Representation of GAIL**). As can be seen from Definition 2, the $\mathcal{R}$-distances are essentially differentiating two policies. For the Wasserstein-type distance, i.e., $\mathcal{R}$ contains all 1-Lipschitz continuous functions, if $d_{\mathcal{R}}(\pi, \pi')$ is small, it is safe to conclude that two policies $\pi$ and $\pi'$ are nearly the same almost everywhere. However, when we choose $\mathcal{R}$ to be the reproducing kernel Hilbert space or the class of neural networks with relatively small complexity, $d_{\mathcal{R}}(\pi, \pi')$ can be small even if $\pi$ and $\pi'$ are not very close. Therefore, we need to choose a sufficiently diverse class of reward functions to ensure that we recover the expert policy.

As Theorem 1 suggests, however, that we need to control the complexity of the function class $\mathcal{R}$ to guarantee the generalization. This implies that when parameterizing the reward function, we need to carefully choose the function class to attain the optimal tradeoff between generalization and representation of GAIL.

## 3   Computation of GAIL

To investigate the computational properties of GAIL, we parameterize the reward by functions belonging to some reproducing kernel Hilbert space. The implementation is based on feature mapping, as mentioned in the previous section. The policy can be parameterized by functions belonging to some reproducing kernel Hilbert space or some class of deep neural networks with parameter $\omega$. Specifically, we denote $\pi(a|s) = \widetilde{\pi}_\omega(\psi_s)$, where $\widetilde{\pi}_\omega(\psi_s)$ is the parametrized policy mapping from $\mathbb{R}^{d_\mathcal{S}}$ to a simplex in $\mathbb{R}_{\mathcal{A}}^d$ with $|\mathcal{A}| = d$. For computational convenience, we consider solving a slightly modified minimax optimization problem:

$$\min_{\omega} \max_{\|\theta\|_2 \leq \kappa} \mathbb{E}_{\widetilde{\pi}_\omega}[\widetilde{r}_\theta(s, a)] - \mathbb{E}_{\pi^*}\widetilde{r}_\theta[(\psi_s, \psi_a)] - \lambda H(\widetilde{\pi}_\omega) - \frac{\mu}{2}\|\theta\|_2^2, \tag{2}$$

where $\widetilde{r}_\theta(s, a) = \theta^\top g(\psi_s, \psi_a)$, $H(\widetilde{\pi}_\omega)$ is some regularizer for the policy (e.g., causal entropy regularizer, Ho & Ermon (2016)), and $\lambda > 0$ and $\mu > 0$ are tuning parameters. Compared with (1), the additional regularizers in (2) can improve the optimization landscape, and help mitigate computational instability in practice.

We apply the alternating mini-batch stochastic gradient algorithm to (2). Specifically, we denote the objective function in (2) as $F(\omega, \theta)$ for notational simplicity. At the $(t + 1)$-th iteration, we take

$$\theta^{(t+1)} = \Pi_\kappa\left(\theta^{(t)} + \frac{\eta_\theta}{q_\theta}\sum_{j \in \mathcal{M}_\theta^{(t)}} \nabla_\theta f_j(\omega^{(t)}, \theta^{(t)})\right) \quad \text{and} \tag{3}$$

$$\omega^{(t+1)} = \omega^{(t)} - \frac{\eta_\omega}{q_\omega}\sum_{j \in \mathcal{M}_\omega^{(t)}} \nabla_\omega \widetilde{f}_j(\omega^{(t)}, \theta^{(t+1)}), \tag{4}$$

where $\eta_\theta$ and $\eta_\omega$ are learning rates, the projection $\Pi_\kappa(v) = \mathbb{1}(\|v\|_2 \leq \kappa) \cdot v + \mathbb{1}(\|v\|_2 > \kappa) \cdot \kappa \cdot v/\|v\|_2$, $\nabla f_j$'s and $\nabla \widetilde{f}_j$'s are independent stochastic approximations of $\nabla F$ (Sutton et al., 2000), and $\mathcal{M}_\theta^{(t)}$, $\mathcal{M}_\omega^{(t)}$ are mini-batches with sizes $q_\theta$ and $q_\omega$, respectively. Before we proceed with the convergence analysis, we impose the follow assumptions on the problem.

**Assumption 4.** There are two positive constants $M_\omega$ and $M_\theta$ such that for any $\omega$ and $\|\theta\|_2 \leq \kappa$,

Unbiased : $\mathbb{E}\nabla f_j(\omega, \theta) = \mathbb{E}\nabla \widetilde{f}_j(\omega, \theta) = \nabla F(\omega, \theta)$,

Bounded : $\mathbb{E}\|\nabla_\omega \widetilde{f}_j(\omega, \theta) - \nabla_\omega F(\omega, \theta)\|_2^2 \leq M_\omega \quad \text{and} \quad \mathbb{E}\|\nabla_\theta f_j(\omega, \theta) - \nabla_\theta F(\omega, \theta)\|_2^2 \leq M_\theta.$

Assumption 4 requires the stochastic gradient to be unbiased with a bounded variance, which is a common assumption in existing optimization literature (Nemirovski et al., 2009; Ghadimi & Lan, 2013; Duchi et al., 2011; Bottou, 2010).

**Assumption 5.** (i) For any $\omega$, there exists some constant $\chi > 0$ and $\upsilon \in (0, 1)$ such that

$$\|(P_{\widetilde{\pi}_\omega})^t \rho_0 - \rho_{\widetilde{\pi}_\omega}\|_{\mathrm{TV}} \le \chi \upsilon^t,$$

where $P_{\widetilde{\pi}_\omega}(s', a' \mid s, a) = \widetilde{\pi}_\omega(a'|s')P(s' \mid s, a)$ is the transition kernel induced by $\widetilde{\pi}_\omega$, $\rho_0$ is the initial distribution of $(s_0, a_0)$, and $\rho_{\widetilde{\pi}_\omega}$ is the stationary distribution induced by $\widetilde{\pi}_\omega$.

(ii) There exist constants $S_{\widetilde{\pi}}, B_\omega, L_\rho, L_Q > 0$ such that for any $\omega, \omega'$, we have

$$\|\nabla_\omega \log(\widetilde{\pi}_\omega(a|s)) - \nabla_\omega \log(\widetilde{\pi}_{\omega'}(a|s))\|_2 \le S_{\widetilde{\pi}} \|\omega - \omega'\|_2, \quad \|\nabla_\omega \log \widetilde{\pi}_\omega(a|s)\|_2 \le B_\omega,$$

$$\|\rho_{\widetilde{\pi}_\omega} - \rho_{\widetilde{\pi}'_\omega}\|_{\mathrm{TV}} \le L_\rho\|\omega - \omega'\|_2, \qquad\qquad \|Q^{\widetilde{\pi}_\omega} - Q^{\widetilde{\pi}_{\omega'}}\|_\infty \le L_Q \|\omega - \omega'\|_2,$$

where $Q^{\widetilde{\pi}_\omega}(s,a) = \sum_{t=0}^\infty \mathbb{E}\left[\widetilde{r}(s_t, a_t) - \mathbb{E}_{\widetilde{\pi}_\omega}[\widetilde{r}] \mid s_0 = s, a_0 = a, \widetilde{\pi}_\omega\right]$ is the action-value function.

(iii) There exist constants $B_H$ and $S_H > 0$ such that for any $\omega, \omega'$, we have

$$H(\widetilde{\pi}_\omega) \le B_H, \quad \text{and} \quad \|\nabla_\omega H(\widetilde{\pi}_\omega) - \nabla_\omega H(\widetilde{\pi}_{\omega'})\|_2 \le S_H \|\omega - \omega'\|_2.$$

Note that (i) of Assumption 5 requires the Markov Chain to be geometrically mixing. (ii) and (iii) state some commonly used regularity conditions for policies (Sutton et al., 2000; Pirotta et al., 2015).

We then define $L$-stationary points of $F$. Specifically, we say that $(\omega^*, \theta^*)$ is a stationary point of $F$, if and only if, for any fixed $\alpha > 0$,

$$\nabla_\omega F(\omega^*, \theta^*) = 0 \text{ and } \theta^* - \Pi_\kappa(\theta^* + \alpha\nabla_\theta F(\omega^*, \theta^*)) = 0.$$

The $L$-stationarity is a generalization of the stationary point for unconstrained optimization, and is a necessary condition for optimality. Accordingly, we take $\alpha = 1$ and measure the sub-stationarity of the algorithm at the iteration $N$ by

$$J_N = \min_{1 \le t \le N} \mathbb{E}\|\theta^{(t)} - \Pi_\kappa(\theta^{(t)} + \nabla_\theta F(\omega^{(t)}, \theta^{(t)}))\|_2^2 + \mathbb{E}\|\nabla_\omega F(\omega^{(t)}, \theta^{(t+1)})\|_2^2.$$

We then state the global convergence of the alternating mini-batch stochastic gradient algorithm.

**Theorem 2.** Suppose Assumptions 1-5 hold. We choose step sizes $\eta_\theta, \eta_\omega$ satisfying

$$\eta_\omega \le \min\left\{\frac{L_\omega}{S_\omega(8L_\omega + 2)}, \frac{1}{2L_\omega}\right\}, \quad \eta_\theta \le \min\left\{\frac{1}{150\mu}, \frac{7L_\omega + 1}{150S_\omega^2}, \frac{1}{100(2\mu + S_\omega)}\right\},$$

and meanwhile $\eta_\omega/\eta_\theta \le \mu/(30L_\omega + 5)$, where $L_\omega = 2\sqrt{2}(S_{\widetilde{\pi}} + 2B_\omega L_\rho)\kappa\rho_g\chi/(1 - \upsilon) + B_\omega L_Q$, and $S_\omega = 2\sqrt{2q}\kappa\rho_g\chi B_\omega/(1 - \upsilon)$. Given any $\epsilon > 0$, we choose batch sizes $q_\theta = \widetilde{O}(1/\epsilon)$ and $q_\omega = \widetilde{O}(1/\epsilon)$. Then we need at most

$$N = \eta(C_0 + 4\sqrt{2}\rho_g\kappa + \mu\kappa^2 + 2\lambda B_H)\epsilon^{-1}$$

iterations such that $J_N \le \epsilon$, where $C_0$ depends on the initialization, and $\eta$ depends on $\eta_\omega$ and $\eta_\theta$.

Here $\widetilde{O}$ hides linear or quardic dependence on some constants in Assumptions 1-5. Theorem 2 shows that though the minimax optimization problem in (2) does not have a convex-concave structure, the alternating mini-batch stochastic gradient algorithm still guarantees to converge to a stationary point. We are not aware of any similar results for GAIL in existing literature.

**Proof Sketch.** We prove the convergence by showing

$$\sum_{i=1}^N \mathbb{E}\|\theta^{(t)} - \Pi_\kappa(\theta^{(t)} + \nabla_\theta F(\omega^{(t)}, \theta^{(t)}))\|_2^2 + \mathbb{E}\|\nabla_\omega F(\omega^{(t)}, \theta^{(t+1)})\|_2^2 \le C + N\epsilon/2, \quad (5)$$

where $C$ is a constant and $N\epsilon/2$ is the accumulation of noise in stochastic approximations of $\nabla F$. Then we straightforwardly have $NJ_N \le C + N\epsilon/2$. Dividing both sizes by $N$, we can derive the desired result. The main difficulty of showing (5) comes from the fact that the outer minimization problem is nonconvex and we cannot solve the inner maximization problem exactly. To overcome this difficulty, we construct a monotonically decreasing potential function:

$$\mathcal{E}^{(t)} = \mathbb{E}F(\omega^{(t)}, \theta^{(t)}) + s\big((1 + 2\eta_\omega L_\omega)/2 \cdot \mathbb{E}\|\omega^{(t)} - \omega^{(t-1)}\|_2^2$$
$$+ (\eta_\omega/2\eta_\theta - \mu\eta_\omega/4 + 3\eta_\omega\eta_\theta\mu^2/2) \cdot \mathbb{E}\|\theta^{(t+1)} - \theta^{(t)}\|_2^2 + \mu\eta_\omega/8 \cdot \mathbb{E}\|\theta^{(t)} - \theta^{(t-1)}\|_2^2\big),$$

for a constant $s$ to be chosen later. Denote $\xi_\theta^{(t)}$ and $\xi_\omega^{(t)}$ as the i.i.d. noise of the stochastic gradients. The following lemma characterizes the decrement of the potential function at each iteration.

**Lemma 1.** With the step sizes $\eta_\theta$ and $\eta_\omega$ chosen as in Theorem 2, we have

$$\mathcal{E}^{(t+1)} - \mathcal{E}^{(t)} \leq - k_1\mathbb{E}\|\omega^{(t+1)} - \omega^{(t)}\|_2^2 - k_2\mathbb{E}\|\omega^{(t)} - \omega^{(t-1)}\|_2^2 - k_3\mathbb{E}\|\theta^{(t+2)} - \theta^{(t+1)}\|_2^2$$
$$- k_4\mathbb{E}\|\theta^{(t+1)} - \theta^{(t)}\|_2^2 - k_5\mathbb{E}\|\theta^{(t)} - \theta^{(t-1)}\|_2^2 + \nu(\mathbb{E}\|\xi_\omega^{(t)}\|_2^2 + \mathbb{E}\|\xi_\theta^{(t)}\|_2^2),$$

where $\nu$ is a constant depending on $F$, $\eta_\theta$, and $\eta_\omega$. Moreover, we have constants $k_1, k_2, k_3, k_4, k_5 > 0$ for $s = 8/(\eta_\omega^2(58L_\omega + 9))$.

Let $k = 1/\min\{k_1, k_4\}$ and $\phi = \max\{1, 1/\eta_\theta^2, 1/\eta_\omega^2\}$. We obtain

$$\sum_{t=1}^N \mathbb{E}\|\theta^{(t)} - \Pi_\kappa(\theta^{(t)} + \nabla_\theta F(\omega^{(t)}, \theta^{(t)}))\|_2^2 + \mathbb{E}\|\nabla_\omega F(\omega^{(t)}, \theta^{(t+1)})\|_2^2$$

$$\overset{(i)}{\leq} \phi \sum_{t=1}^N \mathbb{E}[\|\theta^{(t+1)} - \theta^{(t)}\|_2^2 + \|\omega^{(t+1)} - \omega^{(t)}\|_2^2] \overset{(ii)}{\leq} k\phi(\mathcal{E}^{(1)} - \mathcal{E}^{(N)}) + k\phi N\nu\mathbb{E}[\|\xi_\omega^{(t)}\|_2^2 + \|\xi_\theta^{(t)}\|_2^2],$$

where (i) follows from plugging in the update (3) as well as the contraction property of projection, and (ii) follows from Lemma 1. Choosing $q_\theta = 4k\phi\nu M_\theta/\epsilon$ and $q_\omega = 4k\phi\nu M_\omega/\epsilon$, we obtain

$$\sum_{t=1}^N \mathbb{E}\|\theta^{(t)} - \Pi_\kappa(\theta^{(t)} + \nabla_\theta F(\omega^{(t)}, \theta^{(t)}))\|_2^2 + \mathbb{E}\|\nabla_\omega F(\omega^{(t)}, \theta^{(t+1)})\|_2^2 \leq k\phi(\mathcal{E}^{(1)} - \mathcal{E}^{(N)}) + \frac{N\epsilon}{2}.$$

We have $\mathcal{E}^{(N)} \geq \mathbb{E}F(\omega^{(N)}, \theta^{(N)})$ by the construction of $\mathcal{E}^{(N)}$. It is easy to verify that $F$ is lower bounded (Lemma 10 in Appendix B). Eventually, we complete the proof by substituting the lower bound and choosing $N = k\phi(2\mathcal{E}^{(1)} + 4\sqrt{2}\rho_g\kappa + \mu\kappa^2 + 2\lambda B_H)\epsilon^{-1}$. $\qquad\square$

## 4 EXPERIMENT

To verify our theory in Section 3, we conduct experiments in three reinforcement learning tasks: Acrobot, MountainCar, and Hopper. For each task, we first train an expert policy using the proximal policy optimization (PPO) algorithm in (Schulman et al., 2017) for 500 iterations, and then use the expert policy to generate the demonstration data. The demonstration data for every task contains 500 trajectories, each of which is a series of state action pairs throughout one episode in the environment. When training GAIL, we randomly select a mini-batch of trajectories, which contain at least 8192 state action pairs. We use PPO to update the policy parameters. This avoids the instability of the policy gradient algorithm, and improves the reproducibility of our experiments.

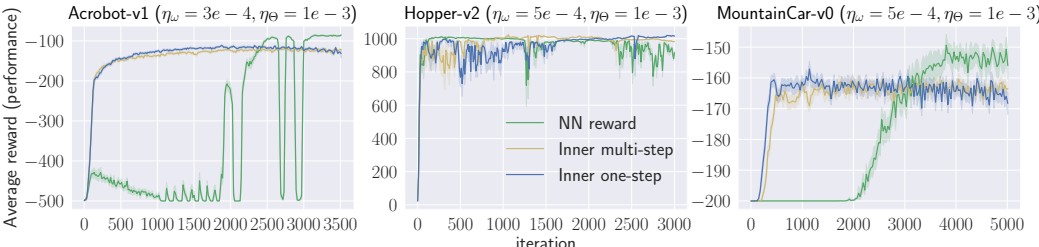

Figure 1: Performance of GAIL on three different tasks. The plotted curves are averaged over 5 independent runs with the vertical axis being the average reward and horizontal axis being the number of iterations.

We use the same neural network architecture for all the environments. For policy, we use a fully connected neural network with two hidden layers of 128 neurons in each layer and $\tanh$ activation. For reward, we use a fully connected ReLU neural network with two hidden layers of 1024 and 512 neurons, respectively. To implement the kernel reward, we fix the first two layers of the neural network after random initialization and only update the third layer, i.e., the first two layers mimic the random feature mapping. We choose $\kappa = 1$ and $\mu = 0.3$. When updating the neural network reward, we use weight normalization in each layer (Salimans & Kingma, 2016).

When updating the kernel reward at each iteration, we choose to take the stochastic gradient ascent step for either once (i.e., alternating update in Section 3) or 10 times. When updating the neural network reward at each iteration, we choose to take the stochastic gradient ascent step for only once. We tune step size parameters for updating the policy and reward, and summarize the numerical results of the step sizes attaining the maximal average episode reward in Figure 1.

As can be seen, using multiple stochastic gradient ascent steps for updating the reward at each iteration yields similar performance as that of one step. We present the convergence analysis of using multiple stochastic gradient ascent steps for updating the reward in Appendix C. Moreover, we observe that parameterizing the reward by neural networks slightly outperform that of the kernel reward. However, its training process tends to be unstable and takes longer time to converge.

## 5 DISCUSSIONS

Our proposed theories of GAIL are closely related to Generative Adversarial Networks (Goodfellow et al., 2014; Arjovsky et al., 2017): (1) The generalization of GANs is defined based on the integral probabilistic metric (IPM) between the synthetic distribution obtained by the generator network and the distribution of the real data (Arora et al., 2017). As the real data in GANs are considered as independent realizations of the underlying distribution, the generalization of GANs can be analyzed using commonly used empirical process techniques for i.i.d. random variables. GAIL, however, involves dependent demonstration data from experts, and therefore the analysis is more involved. (2) Our computational theory of GAIL can be applied to MMD-GAN and its variants, where the IPM is induced by some reproducing kernel Hilbert space (Li et al., 2017; Bińkowski et al., 2018; Arbel et al., 2018). The alternating mini-batch stochastic gradient algorithm attains a similar sublinear rate of convergence to a stationary solution.

Moreover, our computational theory of GAIL only considers the policy gradient update when learning the policy (Sutton et al., 2000). Extending to other types of updates such as natural policy gradient (Kakade, 2002), proximal policy gradient (Schulman et al., 2017) and trust region policy optimization (Schulman et al., 2015) is a challenging, but important future direction.

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

# A  PROOFS IN SECTION 2

## A.1  PROOF OF THEOREM 1

We first consider $n = 1$. For notational simplicity, we denote $x_t = (s_t, a_t)$ and $\tau = \{x_t\}_{t=0}^{T-1}$. Then the generalization gap is bounded by

$$
\begin{aligned}
d_{\mathcal{R}}(\pi^*, \widehat{\pi}) - \inf_\pi d_{\mathcal{R}}(\pi^*, \pi) &= d_{\mathcal{R}}(\pi^*, \widehat{\pi}) - d_{\mathcal{R}}(\pi_n^*, \widehat{\pi}) \\
&\quad + d_{\mathcal{R}}(\pi_n^*, \widehat{\pi}) - \inf_\pi d_{\mathcal{R}}(\pi_n^*, \pi) \\
&\quad + \inf_\pi d_{\mathcal{R}}(\pi_n^*, \pi) - \inf_\pi d_{\mathcal{R}}(\pi^*, \pi) \\
&\leq 2 \left( \sup_{r \in \mathcal{R}} \mathbb{E}_{\pi^*} r(s, a) - \frac{1}{T} \sum_{t=0}^{T-1} r(s_t, a_t) \right) + \epsilon.
\end{aligned}
$$

Denote $\Phi(\tau) = \sup_{r \in \mathcal{R}} \mathbb{E}_{\pi^*} r(s, a) - \frac{1}{T} \sum_{t=0}^{T-1} r(s_t, a_t) = \sup_{r \in \mathcal{R}} \mathbb{E}_{\pi^*} r(x) - \frac{1}{T} \sum_{t=0}^{T-1} r(x_t)$. We utilize the independent block technique first proposed in Yu (1994) to show the concentration of $\Phi(\tau)$. Specifically, we partition $\tau$ into $2m$ blocks of equal size $b$. We denote two alternating sequences as

$$
\begin{aligned}
\tau_0 &= (X_1, X_2, \cdots, X_m) & X_i &= (x_{(2i-1)b+1}, \cdots, x_{(2i-1)b+b}) \\
\tau_1 &= (X_1^{(1)}, X_2^{(1)}, \cdots, X_m^{(1)}) & X_i &= (x_{2ib+1}, \cdots, x_{2ib+b}),
\end{aligned}
$$

We now define a new sequence

$$
\widetilde{\tau}_0 = (\widetilde{X}_1, \widetilde{X}_2, \cdots, \widetilde{X}_m), \tag{6}
$$

where $\widetilde{X}$'s are i.i.d blocks of size $b$, and each block $\widetilde{X}_i$ follows the same distribution as $X_i$.

We define $r_b : X \to \mathbb{R}$ as $r_b(X) = \frac{1}{b} \sum_{k=1}^{b} r(x_k)$. Note that $r_b$ is essentially the average reward on a block. Accordingly, we denote $\mathcal{R}_b$ as the set of all $r_b$'s induced by $r \in \mathcal{R}$.

Before we proceed, we need to introduce a lemma which characterizes the relationship between the expectations of a bounded measurable function with respect to $\tau_0$ and $\widetilde{\tau}_0$.

**Lemma 2.** (Yu, 1994) Suppose $h$ is a measurable function bounded by $M > 0$ defined over the blocks $X_i$, then the following inequality holds:

$$
|\mathbb{E}_{\tau_0}(h) - \mathbb{E}_{\widetilde{\tau}_0}(h)| \leq (m-1)M\beta(b),
$$

where $\mathbb{E}_{\tau_0}$ denotes the expectation with repect to $\tau_0$, and $\mathbb{E}_{\widetilde{\tau}_0}$ denotes the expectation with respect to $\widetilde{\tau}_0$.

**Corollary 3.** Applying Lemma 2, we have

$$
\mathbb{P}_\tau(\Phi(\tau) > \epsilon) \leq 2\mathbb{P}_{\widetilde{\tau}_0}\left(\Phi(\widetilde{\tau}_0) - \mathbb{E}_{\widetilde{\tau}_0}[\Phi(\widetilde{\tau}_0)] > \epsilon - \mathbb{E}_{\widetilde{\tau}_0}[\Phi(\widetilde{\tau}_0)]\right) + 2(m-1)\beta(b). \tag{7}
$$

*Proof.* Consider $\mathbb{P}(\Phi(\tau) > \epsilon)$, we have

$$
\mathbb{P}_\tau(\Phi(\tau) > \epsilon) = \mathbb{P}_\tau(\sup_{r \in \mathcal{R}} \mathbb{E}_{\pi^*} r(x) - \frac{1}{T} \sum_{t=0}^{T} r(x_t) > \epsilon)
$$

$$
\leq \mathbb{P}_\tau \left( \frac{\sup_{r \in \mathcal{R}} \mathbb{E}_{\pi^*} r(x) - \frac{2}{T} \sum_{t \in \tau_0} r(x_t)}{2} + \frac{\sup_{r \in \mathcal{R}} \mathbb{E}_{\pi^*} r(x) - \frac{2}{T} \sum_{t \in \tau_1} r(x_t)}{2} > \epsilon \right) \tag{8}
$$

$$
\begin{aligned}
&= \mathbb{P}_\tau(\Phi(\tau_0) + \Phi(\tau_1) > 2\epsilon) \\
&\leq \mathbb{P}_{\tau_0}(\Phi(\tau_0) > \epsilon) + \mathbb{P}_{\tau_1}(\Phi(\tau_1) > \epsilon) \\
&= 2\mathbb{P}_{\tau_0}(\Phi(\tau_0) > \epsilon) \\
&= 2\mathbb{P}_{\tau_0}\left(\Phi(\tau_0) - \mathbb{E}_{\widetilde{\tau}_0}[\Phi(\widetilde{\tau}_0)] > \epsilon - \mathbb{E}_{\widetilde{\tau}_0}[\Phi(\widetilde{\tau}_0)]\right),
\end{aligned}
$$

where the first inequality (8) follows from the convexity of supremum.

Applying Lemma 2 and setting $h = \mathbb{1}\left\{(\Phi(\widetilde{\tau}_0) - \mathbb{E}_{\widetilde{\tau}_0}[\Phi(\widetilde{\tau}_0)] > \epsilon - \mathbb{E}_{\widetilde{\tau}_0}[\Phi(\widetilde{\tau}_0)]\right\}$, we obtain

$$\mathbb{P}_{\tau_0}(\Phi(\tau_0) - \mathbb{E}_{\widetilde{\tau}_0}[\Phi(\widetilde{\tau}_0)] > \epsilon - \mathbb{E}_{\widetilde{\tau}_0}[\Phi(\widetilde{\tau}_0)])$$
$$\leq \mathbb{P}_{\widetilde{\tau}_0}((\Phi(\widetilde{\tau}_0) - \mathbb{E}_{\widetilde{\tau}_0}[\Phi(\widetilde{\tau}_0)] > \epsilon - \mathbb{E}_{\widetilde{\tau}_0}[\Phi(\widetilde{\tau}_0)]) + 2(m-1)\beta(b).$$

$\square$

Now since $\widetilde{\tau}_0$ consists of independent blocks, we can apply McDiarmid's inequality to $r_b$ by viewing $\widetilde{X}_i$'s as i.i.d samples. We rewrite $\Phi(\widetilde{\tau}_0)$ as

$$\Phi(\widetilde{\tau}_0) = \sup_{r_b \in \mathcal{R}_b} \mathbb{E}_{\pi^*} r_b(\widetilde{X}_i) - \frac{1}{m}\sum_{i=1}^{m} r_b(\widetilde{X}_i).$$

Given samples $\widetilde{X}_1, \cdots, \widetilde{X}_i, \cdots, \widetilde{X}_m$ and $\widetilde{X}_1, \cdots, \widetilde{X}_i', \cdots, \widetilde{X}_m$ , we have

$$\left|\Phi(\widetilde{X}_1, \cdots, \widetilde{X}_i, \cdots, \widetilde{X}_m) - \Phi(\widetilde{X}_1, \cdots, \widetilde{X}_i', \cdots, \widetilde{X}_m)\right| \leq \frac{2}{m}\left|r_b(\widetilde{X}_i)\right| \leq \frac{2B_r}{m}.$$

Then by McDiarmid's inequality, we have

$$\mathbb{P}_{\widetilde{\tau}_0}\left(\Phi(\widetilde{\tau}_0) - \mathbb{E}_{\widetilde{\tau}_0}[\Phi(\widetilde{\tau}_0)] > \epsilon - \mathbb{E}_{\widetilde{\tau}_0}[\Phi(\widetilde{\tau}_0)]\right) \leq \exp\left(\frac{-m(\epsilon - \mathbb{E}_{\widetilde{\tau}_0}[\Phi(\widetilde{\tau}_0)])^2}{2B_r^2}\right). \tag{9}$$

Now combining (7) and (9), we obtain

$$\mathbb{P}_{\tau}(\Phi(\tau) > \epsilon) \leq 2\exp\left(\frac{-m(\epsilon - \mathbb{E}_{\widetilde{\tau}_0}[\Phi(\widetilde{\tau}_0)])^2}{2B_r^2}\right) + 2(m-1)\beta(b). \tag{10}$$

By the argument of symmetrization, we have

$$\mathbb{E}_{\widetilde{\tau}_0}[\Phi(\widetilde{\tau}_0)] \leq 2\mathbb{E}_{\widetilde{\tau}_0,\sigma}\left[\frac{1}{m}\sup_{r_b \in \mathcal{R}_b}\sum_{i=1}^{m}\sigma_i r_b(\widetilde{X}_i)\right], \tag{11}$$

where $\sigma_i$'s are i.i.d. Rademacher random variables. Now we relate the Rademacher complexity (11) to its counterpart taking i.i.d samples. Specifically, we denote $x_j^{(t)}$ as the $j$-th point of the $t$-th block. Denote $\widetilde{\tau}_0^j$ as the collection of the $j$-th sample from each independent block $\widetilde{X}_i$ for $i = 1, \ldots, m$. Plugging in the definition of $r_b$, we have

$$2\mathbb{E}_{\widetilde{\tau}_0,\sigma}\left[\frac{1}{m}\sup_{r \in \mathcal{R}}\sum_{t=1}^{m}\sigma_t \frac{1}{b}\sum_{j=1}^{b}r(x_j^{(t)})\right] \leq 2\mathbb{E}_{\widetilde{\tau}_0,\sigma}\left[\frac{1}{b}\sum_{j=1}^{b}\frac{1}{m}\sup_{r \in R}\sum_{t=1}^{m}\sigma_t r(x_j^{(t)})\right]$$

$$\leq \frac{2}{b}\sum_{j=1}^{b}\mathbb{E}_{\widetilde{\tau}_0,\sigma}\left[\frac{1}{m}\sup_{r \in \mathcal{R}}\sum_{t=1}^{m}\sigma_t r(x_j^{(t)})\right]$$

$$= \frac{2}{b}\sum_{j=1}^{b}\mathbb{E}_{\widetilde{\tau}_0^j,\sigma}\left[\frac{1}{m}\sup_{r \in \mathcal{R}}\sum_{t=1}^{m}\sigma_t r(x_j^{(t)})\right]$$

$$= 2\mathbb{E}_{\widetilde{\tau}_0^1,\sigma}\left[\frac{1}{m}\sup_{r \in \mathcal{R}}\sum_{t=1}^{m}\sigma_t r(x_1^{(t)})\right]. \tag{12}$$

Setting the right-hand side of (10) to be $\frac{\delta}{2}$ and substituting (12), we obtain, with probability at least $1 - \frac{\delta}{2}$, for all $r \in \mathcal{R}$,

$$\Phi(\tau) \leq 2\mathbb{E}_{\widetilde{\tau}_0^1,\sigma}\left[\frac{1}{m}\sup_{r \in \mathcal{R}}\sum_{t=1}^{m}\sigma_t r(x_1^{(t)})\right] + 2B_r\sqrt{\frac{\log\frac{4}{\delta'}}{2m}}, \tag{13}$$

where $\delta' = \delta - 4(m-1)\beta(b)$.

Then we denote

$$\text{Rademacher complexity for } \widetilde{X}_i\text{'s:}\quad \mathfrak{R}_m^{\widetilde{D}} = \mathbb{E}_{\widetilde{\tau}_0^1,\sigma}\left[\frac{1}{m}\sup_{r\in\mathcal{R}}\sum_{t=1}^m \sigma_i r(x_1^{(t)})\right],$$

$$\text{Empirical Rademacher complexity for } \widetilde{X}_i\text{'s:}\quad \widehat{\mathfrak{R}}_{\widetilde{\tau}_0^1} = \mathbb{E}_{\sigma}\left[\frac{1}{m}\sup_{r\in\mathcal{R}}\sum_{t=1}^m \sigma_i r(\widetilde{x}_1^{(t)})\right],$$

$$\text{Empirical Rademacher complexity for } X_i\text{'s:}\quad \widehat{\mathfrak{R}}_m = \mathbb{E}_{\sigma}\left[\frac{1}{m}\sup_{r\in\mathcal{R}}\sum_{t=1}^m \sigma_i r(x_1^{(t)})\right].$$

Applying Lemma 2 to the indicator function $\mathbb{1}\{\mathfrak{R}_m^{\widetilde{D}} - \widehat{\mathfrak{R}}_m > \epsilon\}$, we obtain

$$\mathbb{P}(\mathfrak{R}_m^{\widetilde{D}} - \widehat{\mathfrak{R}}_m > \epsilon) \le \mathbb{P}(\mathfrak{R}_m^{\widetilde{D}} - \widehat{\mathfrak{R}}_{\widetilde{\tau}_0^1} > \epsilon) + (m-1)\beta(2b-1) \le \mathbb{P}(\mathfrak{R}_m^{\widetilde{D}} - \widehat{\mathfrak{R}}_{\widetilde{\tau}_0^1} > \epsilon) + (m-1)\beta(b).$$

It is straightforward to verify that given $x_1^{(1)},\ldots,x_1^{(i)},\ldots,x_1^{(m)}$ and $x_1^{(1)},\ldots,x_1'^{(i)},\ldots,x_1^{(m)}$, the Rademacher complexity satisfies

$$\left|\widehat{\mathfrak{R}}'_{\widetilde{\tau}_0^1} - \widehat{\mathfrak{R}}_{\widetilde{\tau}_0^1}\right| \le \frac{2B_r}{m}.$$

Then by applying McDiarmid's Inequality again, we obtain

$$\mathbb{P}\left(\mathfrak{R}_m^{\widetilde{D}} - \widehat{\mathfrak{R}}_m > \epsilon\right) \le \exp\left(\frac{-m\epsilon^2}{2B_r^2}\right) + (m-1)\beta(b).$$

Thus with probability at least $1 - \frac{\delta}{2}$, we have

$$\mathfrak{R}_m^{\widetilde{D}} - \widehat{\mathfrak{R}}_m \le 2B_r\sqrt{\frac{\log\frac{1}{\delta/2-(m-1)\beta(b)}}{2m}} \le 2B_r\sqrt{\frac{\log\frac{4}{\delta'}}{2m}}. \tag{14}$$

Combining (13) and (14), we have with probability $1 - \delta$,

$$\Phi(\tau) \le 2\widehat{\mathfrak{R}}_m + 6B_r\sqrt{\frac{\log\frac{4}{\delta'}}{2m}}. \tag{15}$$

We apply the Dudley's entropy integral to bound $\widehat{\mathfrak{R}}_m$. Specifically, we have

$$\begin{aligned}
\widehat{\mathfrak{R}}_m &\le \frac{4\alpha}{\sqrt{m}} + \frac{12}{m}\int_\alpha^{\sqrt{m}B_r}\sqrt{\log\mathcal{N}(\mathcal{R},\epsilon,\|\cdot\|_\infty)}d\epsilon\\
&\le \frac{4\alpha}{\sqrt{m}} + \frac{12B_r}{\sqrt{m}}\sqrt{\log\mathcal{N}(\mathcal{R},\alpha,\|\cdot\|_\infty)}.
\end{aligned} \tag{16}$$

It suffices to pick $\alpha = \frac{1}{\sqrt{m}}$. By combining (6), (15), and (16), we have with probability at least $1 - \delta$,

$$d_{\mathcal{R}}(\pi^*,\widehat{\pi}) - \inf_\pi d_{\mathcal{R}}(\pi^*,\pi) \le \frac{16}{m} + \frac{48B_r}{\sqrt{m}}\sqrt{\log\mathcal{N}(\mathcal{R},1/\sqrt{m},\|\cdot\|_\infty)} + 12B_r\sqrt{\frac{\log\frac{4}{\delta'}}{2m}} + \epsilon,$$

where $\delta' = \delta - 4(m-1)\beta(b)$ and $2bm = T$. Substituting $m = T/2b$, we have

$$d_{\mathcal{R}}(\pi^*,\widehat{\pi}) - \inf_\pi d_{\mathcal{R}}(\pi^*,\pi) \le \frac{32b}{T} + \frac{48B_r}{\sqrt{T/2b}}\sqrt{\log\mathcal{N}(\mathcal{R},1/\sqrt{T/2b},\|\cdot\|_\infty)} + 12B_r\sqrt{\frac{\log\frac{4}{\delta'}}{T/b}} + \epsilon. \tag{17}$$

Now we instantiate a choice of $b$ and $m$ for expotentially $\beta$-mixing sequences, where the mixing coefficient $\beta(b) \le \beta_0\exp(-\beta_1 b^\alpha)$ for constants $\beta_0,\beta_1,\alpha > 0$. We set $\delta' > \delta - 4(m-1)\beta(b) = \frac{\delta}{2}$.

By a simple calculation, it is enough to choose $b = (\frac{\log(4\beta_0 T/\delta)}{\beta_1})^{1/\alpha}$. Substituting such a $b$ into (17). We have with probability at least $1 - \delta$:

$$d_{\mathcal{R}}(\pi^*, \widehat{\pi}) - \inf_{\pi} d_{\mathcal{R}}(\pi^*, \pi) \le O\left(\frac{B_r}{\sqrt{T/\zeta}}\sqrt{\log\mathcal{N}(\mathcal{R}, \sqrt{\frac{\zeta}{T}}, \|\cdot\|_{\infty})} + B_r\sqrt{\frac{\log(1/\delta)}{T/\zeta}}\right) + \epsilon,$$

(18)

where $\zeta = (\beta_1^{-1}\log\frac{\beta_0 T}{\delta})^{\frac{1}{\alpha}}$.

When $n > 1$, we concatenate $n$ trajectories to form a sequence of length $nT$, and such a sequence is still exponentially $\beta$-mixing. Applying the same technique, we partition the whole sequence into $2nm$ blocks of equal size $b$. Then with probability at least $1 - \delta$, we have

$$d_{\mathcal{R}}(\pi^*, \widehat{\pi}) - \inf_{\pi} d_{\mathcal{R}}(\pi^*, \pi) \le O\left(\frac{B_r}{\sqrt{nT/\zeta}}\sqrt{\log\mathcal{N}(\mathcal{R}, \sqrt{\frac{\zeta}{nT}}, \|\cdot\|_{\infty})} + B_r\sqrt{\frac{\log(1/\delta)}{nT/\zeta}}\right) + \epsilon.$$

## A.2 PROOF OF COROLLARY 1

The reward function can be bounded by

$$|r(s,a)| = |\theta^{\top} g(\psi_s, \psi_a)| \le \|\theta\|_2 \|g(\psi_s, \psi_a)\|_2 \le \sqrt{2}B_{\theta}\rho_g,$$

where the first inequality comes from Cauchy-Schwartz inequality.

To compute the covering number, we exploit the Lipschitz continuity of $r(s, a)$ with respect to parameter $\theta$. Specifically, for two different parameters $\theta$ and $\theta'$, we have

$$\begin{aligned}
\|r(s,a) - r'(s,a)\|_{\infty} &= \left\|(\theta - \theta')^{\top} g(\psi_s, \psi_a)\right\|_{\infty} \\
&\overset{(i)}{\le} \|\theta - \theta'\|_2 \sup_{(s,a)\in\mathcal{S}\times\mathcal{A}} \|g(\psi_s, \psi_a)\|_2 \\
&\overset{(ii)}{\le} \sqrt{2}\|\theta - \theta'\|_2 \rho_g \sup_{(s,a)\in\mathcal{S}\times\mathcal{A}} \sqrt{\|\psi_s\|_2^2 + \|\psi_a\|_2^2} \overset{(iii)}{\le} \sqrt{2}\rho_g \|\theta - \theta'\|_2,
\end{aligned}$$

where (i) comes from Cauchy-Schwartz inequality, (ii) comes from the Lipschitz continuity of $g$, and (iii) comes from the boundedness of $\psi_s$ and $\psi_a$.

Denote $\Theta = \{\theta \in \mathbb{R}^q : \|\theta\|_2 \le B_{\theta}\}$. By the standard argument of the volume ratio, we have

$$\mathcal{N}(\Theta, \epsilon, \|\cdot\|_2) \le \left(1 + \frac{2B_{\theta}}{\epsilon}\right)^q.$$

Accordingly, we have

$$\begin{aligned}
\mathcal{N}\left(R, \sqrt{\frac{2(\beta_1^{-1}\log(4\beta_0 T/\delta))^{\frac{1}{\chi}}}{T}}, \|\cdot\|_{\infty}\right) &\le \mathcal{N}\left(\Theta, \frac{1}{\sqrt{2}\rho_g}\sqrt{\frac{2(\beta_1^{-1}\log(4\beta_0 T/\delta))^{\frac{1}{\chi}}}{T}}, \|\cdot\|_2\right) \\
&\le \left(1 + 2\sqrt{2}\rho_g B_{\theta}\sqrt{\frac{T}{2(\beta_1^{-1}\log(4\beta_0 T/\delta))^{\frac{1}{\chi}}}}\right)^q. \quad (19)
\end{aligned}$$

Plugging (19) into (18), we have

$$\begin{aligned}
&d_{\mathcal{R}}(\pi^*, \widehat{\pi}) - \inf_{\pi} d_{\mathcal{R}}(\pi^*, \pi) \\
&= O\left(\frac{\rho_g B_{\theta}}{\sqrt{T/\zeta}}\sqrt{q\log\left(\rho_g B_{\theta}\sqrt{\frac{T}{\zeta}}\right)} + \rho_g B_{\theta}\sqrt{\frac{\log(1/\delta)}{T/\zeta}}\right) + \epsilon
\end{aligned}$$

hold, with probability at least $1 - \delta$.

### A.3 Proof of Corollary 2

We investigate the Lipschitz continuity of $r$ with respect to the weight matrices $W_1, \cdots, W_D$. Specifically, given two different sets of matrices $W_1, \cdots, W_D$ and $W_1', \cdots, W_D'$, we have

$$\|r(s,a) - r'(s,a)\|_\infty$$

$$\leq \left\| W_D^\top \sigma(W_{D-1}\sigma(...\sigma(W_1[\psi_a^\top, \psi_s^\top]^\top)...)) - (W_D')^\top \sigma(W_{D-1}'\sigma(...\sigma(W_1'[\psi_a^\top, \psi_s^\top]^\top)...)) \right\|_2$$

$$\leq \left\| W_D^\top \sigma(W_{D-1}\sigma(...\sigma(W_1[\psi_a^\top, \psi_s^\top]^\top)...)) - (W_D')^\top \sigma(W_{D-1}\sigma(...\sigma(W_1[\psi_a^\top, \psi_s^\top]^\top)...)) \right\|_2$$

$$\quad + \left\| (W_D')^\top \sigma(W_{D-1}\sigma(...\sigma(W_1[\psi_a^\top, \psi_s^\top]^\top)...)) - (W_D')^\top \sigma(W_{D-1}'\sigma(...\sigma(W_1'[\psi_a^\top, \psi_s^\top]^\top)...)) \right\|_2$$

$$\leq \|W_D - W_D'\|_2 \left\| \sigma(W_{D-1}\sigma(...\sigma(W_1[\psi_a^\top, \psi_s^\top]^\top)...)) \right\|_2$$

$$\quad + \|W_D'\|_2 \left\| \sigma(W_{D-1}\sigma(...\sigma(W_1[\psi_a^\top, \psi_s^\top]^\top)...)) - \sigma(W_{D-1}'\sigma(...\sigma(W_1'[\psi_a^\top, \psi_s^\top]^\top)...)) \right\|_2 .$$

Note that we have

$$\left\| \sigma(W_{D-1}\sigma(...\sigma(W_1[\psi_a^\top, \psi_s^\top]^\top)...)) \right\|_2 \overset{(i)}{\leq} \left\| W_{D-1}\sigma(...\sigma(W_1[\psi_a^\top, \psi_s^\top]^\top)...) \right\|_2$$

$$\leq \|W_{D-1}\|_2 \left\| \sigma(...\sigma(W_1[\psi_a^\top, \psi_s^\top]^\top)...) \right\|_2 \overset{(ii)}{\leq} \left\| [\psi_a^\top, \psi_s^\top]^\top \right\|_2 \overset{(iii)}{\leq} \sqrt{2},$$

where (i) comes from the definition of the ReLU activation, (ii) comes from $\|W_i\|_2 \leq 1$ and recursion, and (iii) comes from the boundedness of $\psi_s$ and $\psi_a$. Accordingly, we have

$$\|r(s,a) - r'(s,a)\|_\infty \leq \sqrt{2} \|W_D - W_D'\|_2 + \|W_D'\|_2 \left\| \sigma(W_{D-1}\sigma(...) - \sigma(W_{D-1}'\sigma(...) \right\|_2$$

$$\overset{(i)}{\leq} \sqrt{2} \|W_D - W_D'\|_2 + \left\| W_{D-1}\sigma(...) - W_{D-1}'\sigma(...) \right\|_2$$

$$\overset{(ii)}{\leq} \sum_{i=1}^{D} \sqrt{2} \|W_i - W_i'\|_2 ,$$

where (i) comes from the Lipschitz continuity of the ReLU activation, and (ii) comes from the recursion. We then derive the covering number of $\mathcal{R}$ by the Cartesian product of the matrix covering of $W_1, ..., W_D$:

$$\mathcal{N}(\mathcal{R}, \epsilon, \|\cdot\|_\infty) \leq \prod_{i=1}^{D} \mathcal{N}\left(W_i, \frac{\epsilon}{D\sqrt{2}}, \|\cdot\|_2\right) \leq \left(1 + \frac{\sqrt{2}D\sqrt{d}}{\epsilon}\right)^{d^2 D}, \tag{20}$$

where the second inequality comes from the standard argument of the volume ratio. Plugging (20) into (18), we have

$$d_\mathcal{R}(\pi^*, \widehat{\pi}) - \inf_\pi d_\mathcal{R}(\pi^*, \pi)$$

$$= O\left( \frac{1}{\sqrt{T/\zeta}} \sqrt{d^2 D \log\left(D\sqrt{\frac{dT}{\zeta}}\right)} + \sqrt{\frac{\log(1/\delta)}{T/\zeta}} \right) + \epsilon$$

hold, with probability at least $1 - \delta$.

## B  Proof of Theorem 2

The proof is arranged as follows. We first prove the bounded ness of $Q$ fucntion and characterize the Lipschitz properties of the gradients of $F$ with respect to $\omega$ and $\theta$, respectively, in Section B.2. Then we provide some important lemmas in Section B.3. Using these lemmas, we prove Lemma 1 in Section B.4. We prove Theorem 2 in Section B.5. For notational simplicity, we denote $\langle \cdot, \cdot \rangle$ as the vector inner product throughout the rest of our analysis.

### B.1  Boundedness of $Q$ function

**Lemma 3.** For any $\omega$, we have

$$\left\| Q^{\widetilde{\pi}_\omega} \right\|_\infty \leq B_Q,$$

where $B_Q = \frac{2\sqrt{2}\kappa\rho_g\chi}{1-v}$.

*Proof.*

$$Q^{\widetilde{\pi}_\omega}(s,a) = \sum_{t=0}^{\infty} \mathbb{E}[\widetilde{r}_\theta(s_t,a_t) - \mathbb{E}_{\widetilde{\pi}_\omega}\widetilde{r}_\theta \,|\, s_0 = s, a_0 = a, \widetilde{\pi}_\omega]$$

$$= \sum_{t=0}^{\infty} \Big[ \int_{\mathcal{S}\times\mathcal{A}} \widetilde{r}_\theta(s,a)\rho_0(s,a)(P_{\pi_\omega})^t \mathrm{d}(s,a) - \int_{\mathcal{S}\times\mathcal{A}} \widetilde{r}_\theta(s,a)\rho_{\widetilde{\pi}_\omega}(s,a)\mathrm{d}(s,a) \Big]$$

$$= \sum_{t=0}^{\infty} \int_{\mathcal{S}\times\mathcal{A}} \widetilde{r}_\theta(s,a)[\rho_0(s,a)(P_{\pi_\omega})^t - \rho_{\widetilde{\pi}_\omega}(s,a)]\mathrm{d}(s,a)$$

$$\leq \sum_{t=0}^{\infty} 2\,\|\widetilde{r}_\theta\|_\infty \,\|\rho_0(P_{\pi_\omega})^t - \rho_{\widetilde{\pi}_\omega}\|_{TV}$$

$$\leq 2\sqrt{2}\kappa\rho_g \sum_{t=0}^{\infty} \chi \upsilon^t = \frac{2\sqrt{2}\kappa\rho_g\chi}{1-\upsilon},$$

where the first inequality comes from the definition of Total Variance distance of probability measures and the second inequality results from (i) of Assumption 5. $\qquad\square$

## B.2 Lipschitz properties of the gradients

**Lemma 4.** Suppose that Assumption 1, 3 and 5 hold. For any $\omega, \omega', \theta$ and $\theta'$, we have

$$\|\nabla_\omega F(\omega,\theta) - \nabla_\omega F(\omega',\theta)\|_2 \leq L_\omega \|\omega - \omega'\|_2,$$
$$\|\nabla_\theta F(\omega,\theta) - \nabla_\theta F(\omega,\theta')\|_2 \leq \mu \|\theta - \theta'\|_2,$$

where $L_\omega = \frac{2\sqrt{2}(S_{\widetilde{\pi}}+2B_\omega L_\rho)\kappa\rho_g\chi}{1-\upsilon} + B_\omega L_Q$.

*Proof.* By the Policy Gradient Theorem (Sutton et al., 2000), we have

$$\nabla_\omega F(\omega,\theta) = \mathbb{E}_{\widetilde{\pi}_\omega} \nabla\log(\widetilde{\pi}_\omega(a\,|\,s))Q^{\widetilde{\pi}_\omega}(s,a).$$

Therefore,

$$\|\nabla_\omega F(\omega,\theta) - \nabla_\omega F(\omega',\theta)\|_2$$

$$= \Big\|\mathbb{E}_{\widetilde{\pi}_\omega} \nabla\log(\widetilde{\pi}_\omega(a\,|\,s))Q^{\widetilde{\pi}_\omega}(s,a) - \mathbb{E}_{\widetilde{\pi}_{\omega'}} \nabla\log(\widetilde{\pi}_{\omega'}(a\,|\,s))Q^{\widetilde{\pi}_{\omega'}}(s,a)\Big\|_2$$

$$\leq \Big\|\mathbb{E}_{\widetilde{\pi}_\omega} \nabla\log(\widetilde{\pi}_\omega(a\,|\,s))Q^{\widetilde{\pi}_\omega}(s,a) - \mathbb{E}_{\widetilde{\pi}_\omega} \nabla\log(\widetilde{\pi}_{\omega'}(a\,|\,s))Q^{\widetilde{\pi}_{\omega'}}(s,a)\Big\|_2$$

$$+ \Big\|\mathbb{E}_{\widetilde{\pi}_\omega} \nabla\log(\widetilde{\pi}_{\omega'}(a\,|\,s))Q^{\widetilde{\pi}_{\omega'}}(s,a) - \mathbb{E}_{\widetilde{\pi}_{\omega'}} \nabla\log(\widetilde{\pi}_{\omega'}(a\,|\,s))Q^{\widetilde{\pi}_{\omega'}}(s,a)\Big\|_2$$

$$\leq (S_{\widetilde{\pi}}B_Q + B_\omega L_Q)\|\omega - \omega'\|_2 + 2B_\omega B_Q \|\rho_{\widetilde{\pi}_\omega} - \rho_{\widetilde{\pi}_{\omega'}}\|_{TV}$$

$$\leq (S_{\widetilde{\pi}}B_Q + B_\omega L_Q + 2B_\omega B_Q L_\rho)\|\omega - \omega'\|_2, \tag{21}$$

where the second and the third inequality results from (ii) of Assumption 5. Plugging $B_Q = \frac{2\sqrt{2}\kappa\rho_g\chi}{1-\upsilon}$ into (21) yields the desired result.

Similarly, we have

$$\|\nabla_\theta F(\omega,\theta) - \nabla_\theta F(\omega,\theta')\|_2 \leq \|-\mu(\theta-\theta')\|_2 \leq \mu\|\theta - \theta'\|_2.$$

$\qquad\square$

We then characterize the Lipschitz continuity of $\nabla_\theta F$ with respect to $\omega$.

**Lemma 5.** Suppose Assumptions 1, 3 and 5 hold. For any $\omega, \omega'$ and $\theta$, we have

$$\|\nabla_\theta F(\omega,\theta) - \nabla_\theta F(\omega',\theta)\|_2 \leq S_\omega\|\omega - \omega'\|_2,$$

where $S_\omega = \frac{2\sqrt{2q}\kappa\rho_g\chi B_\omega}{1-\upsilon}$.

*Proof.* We have

$$\nabla_\theta F(\omega, \theta) = -\mu\theta - \nabla_\theta \Big[ \mathbb{E}_{\rho_{\widetilde\pi_\omega}} \big[ \theta^\top g(\psi_{s_t}, \psi_{a_t}) \big] - \mathbb{E}_{\rho^*} \big[ \theta^\top g(\psi_{s_t}, \psi_{a_t}) \big] \Big]$$

$$= -\mu\theta - \Big[ \mathbb{E}_{\rho_{\widetilde\pi_\omega}} \big[ g(\psi_{s_t}, \psi_{a_t}) \big] - \mathbb{E}_{\rho^*} \big[ g(\psi_{s_t}, \psi_{a_t}) \big] \Big].$$

Therefore, we have

$$\Big\| \nabla_\theta F(\omega, \theta) - \nabla_\theta F(\omega', \theta) \Big\|_2$$

$$= \Big\| \mathbb{E}_{\rho_{\widetilde\pi_\omega}} \big[ g(\psi_{s_t}, \psi_{a_t}) \big] - \mathbb{E}_{\rho_{\widetilde\pi_{\omega'}}} \big[ g(\psi_{s_t}, \psi_{a_t}) \big] \Big\|_2$$

$$\leq \sqrt{q} \max_{1 \leq j \leq q} \Big| \mathbb{E}_{\rho_{\widetilde\pi_\omega}} g(\psi_{s_t}, \psi_{a_t})_j - \mathbb{E}_{\rho_{\widetilde\pi_{\omega'}}} g(\psi_{s_t}, \psi_{a_t})_j \Big|. \tag{22}$$

Suppose $j^* = \mathrm{argmax}_{1 \leq j \leq q} \Big| \mathbb{E}_{\rho_{\widetilde\pi_\omega}} g(\psi_{s_t}, \psi_{a_t})_j - \mathbb{E}_{\rho_{\widetilde\pi_{\omega'}}} g(\psi_{s_t}, \psi_{a_t})_j \Big|$, by Mean Value Theorem, there exists vector $\widetilde\omega$, which is some interpolation between vectors $\omega$ and $\omega'$, such that

$$\mathbb{E}_{\rho_{\widetilde\pi_\omega}} g(\psi_{s_t}, \psi_{a_t})_j - \mathbb{E}_{\rho_{\widetilde\pi_{\omega'}}} g(\psi_{s_t}, \psi_{a_t})_j = \langle \nabla_\omega \mathbb{E}_{\rho_{\widetilde\pi_{\widetilde\omega}}} g(\psi_{s_t}, \psi_{a_t})_j, \omega - \omega' \rangle. \tag{23}$$

By Policy Gradient Theorem, we have

$$\Big\| \nabla_\omega \mathbb{E}_{\rho_{\widetilde\pi_{\widetilde\omega}}} g(\psi_{s_t}, \psi_{a_t})_j \Big\|_2 = \Big\| \mathbb{E}_{\rho_{\widetilde\pi_{\widetilde\omega}}} \nabla \log \widetilde\pi_{\widetilde\omega}(a \mid s) Q_g^{\widetilde\pi_{\widetilde\omega}}(s, a) \Big\|_2$$

$$\leq \sup_{(s,a) \in \mathcal{S} \times \mathcal{A}} |\nabla \log \widetilde\pi_{\widetilde\omega}(a \mid s)| |Q_g^{\widetilde\pi_{\widetilde\omega}}(s, a)|$$

$$\leq B_Q B_\omega, \tag{24}$$

where $Q_g^{\widetilde\pi_{\widetilde\omega}}(s, a) = \sum_{t=0}^\infty \mathbb{E}[g(s_t, a_t)_{j^*} - \mathbb{E}_{\widetilde\pi_{\widetilde\omega}} g_{j^*} \mid s_0 = s, a_0 = a, \widetilde\pi_{\widetilde\omega}]$. Combining (22), (23), (24) and using Cauchy-Schwartz Inequality, we prove the lemma. $\qquad\square$

### B.3 SOME IMPORTANT LEMMAS FOR PROVING LEMMA 1

We denote

$$\xi_\theta^{(t)} = \nabla_\theta F(\omega^{(t)}, \theta^{(t)}) - \frac{1}{q_\theta} \sum_{j \in \mathcal{M}_\theta^{(t)}} \nabla_\theta f_j(\omega^{(t)}, \theta^{(t)})$$

$$\text{and } \xi_\omega^{(t)} = \nabla_\omega F(\omega^{(t)}, \theta^{(t+1)}) - \frac{1}{q_\omega} \sum_{j \in \mathcal{M}_\omega^{(t)}} \nabla_\omega f_j(\omega^{(t)}, \theta^{(t+1)})$$

as the i.i.d. noise of the stochastic gradient, respectively. Throughout the rest of the analysis, the expectation $\mathbb{E}$ is taken with respect to all the noise in each iteration of the alternating mini-batch stochastic gradient descent algorithm. The next lemma characterizes the progress at the $(t + 1)$-th iteration. For notational simplicity, we define vector function

$$G(\pi) = \mathbb{E}_{\rho_\pi} g(\psi_s, \psi_a) \tag{25}$$

**Lemma 6.** At the $(t + 1)$-th iteration, we have

$$\mathbb{E}F(\omega^{(t+1)}, \theta^{(t+1)}) - \mathbb{E}F(\omega^{(t)}, \theta^{(t)})$$

$$\leq \Big( L_\omega - \frac{1}{\eta_\omega} \Big) \mathbb{E}\|\omega^{(t+1)} - \omega^{(t)}\|_2^2 + \frac{S_\omega}{2} \cdot \mathbb{E} \Big\| \omega^{(t)} - \omega^{(t-1)} \Big\|_2^2$$

$$+ \Big( \frac{1}{2\eta_\theta} + \frac{S_\omega}{2} + \mu \Big) \mathbb{E} \Big\| \theta^{(t+1)} - \theta^{(t)} \Big\|_2^2 + \Big( \frac{1}{2\eta_\theta} + \frac{\mu}{2} \Big) \mathbb{E} \Big\| \theta^{(t)} - \theta^{(t-1)} \Big\|_2^2$$

$$+ \eta_\omega \mathbb{E} \Big\| \xi_\omega^{(t)} \Big\|_2^2 + \frac{1}{2\mu} \mathbb{E} \Big\| \xi_\theta^{(t-1)} \Big\|_2^2.$$

*Proof.* We have

$$\mathbb{E}F(\omega^{(t+1)}, \theta^{(t+1)}) - \mathbb{E}F(\omega^{(t)}, \theta^{(t)})$$

$$= \mathbb{E}F(\omega^{(t+1)}, \theta^{(t+1)}) - \mathbb{E}F(\omega^{(t)}, \theta^{(t+1)}) + \mathbb{E}F(\omega^{(t)}, \theta^{(t+1)}) - \mathbb{E}F(\omega^{(t)}, \theta^{(t)}).$$

By the mean value theorem, we have

$$\langle \nabla_\omega F(\widetilde{\omega}^{(t)}, \theta^{(t+1)}), \omega^{(t+1)} - \omega^{(t)} \rangle = F(\omega^{(t+1)}, \theta^{(t+1)}) - F(\omega^{(t)}, \theta^{(t+1)}),$$

where $\widetilde{\omega}^{(t)}$ is some interpolation between $\omega^{(t+1)}$ and $\omega^{(t)}$. Then we have

$$
\begin{aligned}
\mathbb{E}F(\omega^{(t+1)}, \theta^{(t+1)}) &- \mathbb{E}F(\omega^{(t)}, \theta^{(t+1)}) \\
&= \mathbb{E}\langle \nabla_\omega F(\widetilde{\omega}^{(t)}, \theta^{(t+1)}) - \nabla_\omega F(\omega^{(t)}, \theta^{(t+1)}), \omega^{(t+1)} - \omega^{(t)} \rangle \\
&\quad + \mathbb{E}\langle \nabla_\omega F(\omega^{(t)}, \theta^{(t+1)}), \omega^{(t+1)} - \omega^{(t)} \rangle.
\end{aligned}
\tag{26}
$$

By Cauchy- Swartz inequality, we have

$$
\begin{aligned}
\mathbb{E}\langle \nabla_\omega F(\widetilde{\omega}^{(t)}, \theta^{(t+1)}) &- \nabla_\omega F(\omega^{(t)}, \theta^{(t+1)}), \omega^{(t+1)} - \omega^{(t)} \rangle \\
&\leq \mathbb{E}\|\nabla_\omega F(\widetilde{\omega}^{(t)}, \theta^{(t+1)}) - \nabla_\omega F(\omega^{(t)}, \theta^{(t+1)})\|_2 \|\omega^{(t+1)} - \omega^{(t)}\|_2 \\
&\leq L_\omega \mathbb{E}\|\omega^{(t+1)} - \omega^{(t)}\|_2^2,
\end{aligned}
$$

where the last inequality comes from Lemma 5. Morever, (4) implies

$$\omega^{(t+1)} - \omega^{(t)} = -\eta_\omega (\nabla_\omega F(\omega^{(t)}, \theta^{(t+1)}) + \xi_\omega^{(t)}).$$

Therefore, we have

$$
\begin{aligned}
\mathbb{E}\langle \nabla_\omega F(\omega^{(t)}, \theta^{(t+1)}), \omega^{(t+1)} - \omega^{(t)} \rangle &= -\frac{1}{\eta_\omega}\mathbb{E}\|\omega^{(t+1)} - \omega^{(t)}\|_2^2 - \mathbb{E}\langle \xi_\omega^{(t)}, \omega^{(t+1)} - \omega^{(t)} \rangle \\
&= -\mathbb{E}\langle \xi_\omega^{(t)}, -\eta_\omega(\nabla_\omega F(\omega^{(t)}, \theta^{(t+1)}) + \xi_\omega^{(t)}) \rangle \\
&\quad -\frac{1}{\eta_\omega}\mathbb{E}\|\omega^{(t+1)} - \omega^{(t)}\|_2^2 \\
&= -\frac{1}{\eta_\omega}\mathbb{E}\|\omega^{(t+1)} - \omega^{(t)}\|_2^2 + \eta_\omega \mathbb{E}\left\|\xi_\omega^{(t)}\right\|_2^2.
\end{aligned}
$$

Thus, we have

$$\mathbb{E}F(\omega^{(t+1)}, \theta^{(t+1)}) - \mathbb{E}F(\omega^{(t)}, \theta^{(t+1)}) \leq (L_\omega - \frac{1}{\eta_\omega})\mathbb{E}\|\omega^{(t+1)} - \omega^{(t)}\|_2^2 + \eta_\omega \mathbb{E}\left\|\xi_\omega^{(t)}\right\|_2^2. \tag{27}$$

By (3), the increment of $F(\omega, \theta)$ takes the form

$$
\begin{aligned}
F(\omega^{(t)}, \theta^{(t+1)}) &- F(\omega^{(t)}, \theta^{(t)}) \\
&= \left\langle G(\widetilde{\pi}_{\omega^{(t)}}) - G(\pi^*), \theta^{(t+1)} - \theta^{(t)} \right\rangle - \frac{\mu}{2}(\|\theta^{(t+1)}\|_2^2 - \|\theta^{(t)}\|_2^2) \\
&\leq \left\langle G(\widetilde{\pi}_{\omega^{(t)}}) - G(\pi^*) - \mu\theta^{(t)}, \theta^{(t+1)} - \theta^{(t)} \right\rangle.
\end{aligned}
\tag{28}
$$

For notational simplicity, we define

$$\epsilon^{(t+1)} = \theta^{(t+1)} - \left(\theta^{(t)} + \eta_\theta\left(\nabla_\theta F(\omega^{(t)}, \theta^{(t)}) + \xi_\theta^{(t)}\right)\right). \tag{29}$$

Note that we have

$$\nabla_\theta F(\omega^{(t)}, \theta^{(t)}) = G(\widetilde{\pi}_{\omega^{(t)}}) - G(\pi^*) - \mu\theta^{(t)}. \tag{30}$$

Plugging (29) and (30) into (28), we obtain

$$F(\omega^{(t)}, \theta^{(t+1)}) - F(\omega^{(t)}, \theta^{(t)}) \leq \left\langle \frac{\theta^{(t+1)} - \theta^{(t)} - \epsilon^{(t+1)}}{\eta_\theta} - \xi_\theta^{(t)}, \theta^{(t+1)} - \theta^{(t)} \right\rangle.$$

Since $\theta$ belongs to the convex set $\{\theta \mid \|\theta\|_2 \leq \kappa\}$, we have

$$\langle \epsilon^{(t)}, \theta^{(t+1)} - \theta^{(t)} \rangle \geq 0.$$

Then we obtain

$$
F(\omega^{(t)}, \theta^{(t+1)}) - F(\omega^{(t)}, \theta^{(t)})
$$
$$
\leq \frac{1}{\eta_\theta} \langle \theta^{(t+1)} - \theta^{(t)} + \epsilon^{(t)} - \epsilon^{(t+1)}, \theta^{(t+1)} - \theta^{(t)} \rangle - \langle \xi_\theta^{(t)}, \theta^{(t+1)} - \theta^{(t)} \rangle
$$
$$
= \frac{1}{\eta_\theta} \langle \epsilon^{(t)} - \epsilon^{(t+1)}, \theta^{(t+1)} - \theta^{(t)} \rangle + \frac{1}{\eta_\theta} \left\| \theta^{(t+1)} - \theta^{(t)} \right\|_2^2 - \langle \xi_\theta^{(t)}, \theta^{(t+1)} - \theta^{(t)} \rangle.
$$
$$(31)$$

By the definition of $\epsilon^{(t)}$ in (29), we have

$$
\epsilon^{(t+1)} = \theta^{(t+1)} - \left( \theta^{(t)} + \eta_\theta \left( \nabla_\theta F(\omega^{(t)}, \theta^{(t)}) + \xi_\theta^{(t)} \right) \right)
$$
$$(32)$$

and

$$
\epsilon^{(t)} = \theta^{(t)} - \left( \theta^{(t-1)} + \eta_\theta \left( \nabla_\theta F(\omega^{(t-1)}, \theta^{(t-1)}) + \xi_\theta^{(t-1)} \right) \right).
$$
$$(33)$$

Subtracting (33) from (32),we obtain

$$
\epsilon^{(t)} - \epsilon^{(t+1)} = (\theta^{(t)} - \theta^{(t+1)}) - (\theta^{(t-1)} - \theta^{(t)}) - \eta_\theta \left( \nabla_\theta F(\omega^{(t-1)}, \theta^{(t-1)}) - \nabla_\theta F(\omega^{(t)}, \theta^{(t)}) \right)
$$
$$
- \eta_\theta (\xi_\theta^{(t-1)} - \xi_\theta^{(t)}).
$$
$$(34)$$

Plugging (34) into the first term on the right hand side of (31), we obtain

$$
\frac{1}{\eta_\theta} \langle \epsilon^{(t)} - \epsilon^{(t+1)}, \theta^{(t+1)} - \theta^{(t)} \rangle
$$
$$
= \underbrace{\frac{1}{\eta_\theta} \langle \theta^{(t)} - \theta^{(t-1)}, \theta^{(t+1)} - \theta^{(t)} \rangle}_{(A)} + \underbrace{\langle \nabla_\theta F(\omega^{(t)}, \theta^{(t)}) - \nabla_\theta F(\omega^{(t-1)}, \theta^{(t-1)}), \theta^{(t+1)} - \theta^{(t)} \rangle}_{(B)}
$$
$$
- \frac{1}{\eta_\theta} \left\| \theta^{(t+1)} - \theta^{(t)} \right\|_2^2 + \langle \xi_\theta^{(t)} - \xi_\theta^{(t-1)}, \theta^{(t+1)} - \theta^{(t)} \rangle.
$$
$$(35)$$

For term $(A)$, we apply the Cauchy-Schwarz inequality to obtain an upper bound as follows.

$$
\frac{1}{\eta_\theta} \langle \theta^{(t)} - \theta^{(t-1)}, \theta^{(t+1)} - \theta^{(t)} \rangle \leq \frac{1}{\eta_\theta} \left\| \theta^{(t)} - \theta^{(t-1)} \right\|_2 \cdot \left\| \theta^{(t+1)} - \theta^{(t)} \right\|_2
$$
$$
\leq \frac{1}{2\eta_\theta} \left\| \theta^{(t)} - \theta^{(t-1)} \right\|_2^2 + \frac{1}{2\eta_\theta} \left\| \theta^{(t+1)} - \theta^{(t)} \right\|_2^2. \quad (36)
$$

To derive the upper bound of $(B)$, we apply Lemma 5 to obtain

$$
\langle \nabla_\theta F(\omega^{(t)}, \theta^{(t)}) - \nabla_\theta F(\omega^{(t-1)}, \theta^{(t-1)}), \theta^{(t+1)} - \theta^{(t)} \rangle
$$
$$
= \langle \nabla_\theta F(\omega^{(t)}, \theta^{(t)}) - \nabla_\theta F(\omega^{(t-1)}, \theta^{(t)}), \theta^{(t+1)} - \theta^{(t)} \rangle
$$
$$
+ \langle \nabla_\theta F(\omega^{(t-1)}, \theta^{(t)}) - \nabla_\theta F(\omega^{(t-1)}, \theta^{(t-1)}), \theta^{(t+1)} - \theta^{(t)} \rangle
$$
$$
\leq S_\omega \left\| \omega^{(t)} - \omega^{(t-1)} \right\|_2 \cdot \left\| \theta^{(t+1)} - \theta^{(t)} \right\|_2 + \mu \cdot \left\| \theta^{(t)} - \theta^{(t-1)} \right\|_2 \cdot \left\| \theta^{(t+1)} - \theta^{(t)} \right\|_2
$$
$$
\leq \frac{S_\omega}{2} \left\| \omega^{(t)} - \omega^{(t-1)} \right\|_2^2 + \frac{S_\omega}{2} \cdot \left\| \theta^{(t+1)} - \theta^{(t)} \right\|_2^2
$$
$$
+ \frac{\mu}{2} \cdot \left\| \theta^{(t)} - \theta^{(t-1)} \right\|_2^2 + \frac{\mu}{2} \cdot \left\| \theta^{(t+1)} - \theta^{(t)} \right\|_2^2.
$$
$$(37)$$

Plugging (36) and (37) into (35), we obtain

$$
\frac{1}{\eta_\theta} \langle \epsilon^{(t)} - \epsilon^{(t+1)}, \theta^{(t+1)} - \theta^{(t)} \rangle \leq \left( -\frac{1}{2\eta_\theta} + \frac{\mu}{2} + \frac{S_\omega}{2} \right) \left\| \theta^{(t+1)} - \theta^{(t)} \right\|_2^2
$$
$$
+ \left( \frac{1}{2\eta_\theta} + \frac{\mu}{2} \right) \left\| \theta^{(t)} - \theta^{(t-1)} \right\|_2^2
$$
$$
+ \frac{S_\omega}{2} \left\| \omega^{(t)} - \omega^{(t-1)} \right\|_2^2 + \langle \xi_\theta^{(t)} - \xi_\theta^{(t-1)}, \theta^{(t+1)} - \theta^{(t)} \rangle. \quad (38)
$$

Further plugging (38) into (31), we obtain

$$
\begin{aligned}
F(\omega^{(t)}, &\theta^{(t+1)}) - F(\omega^{(t)}, \theta^{(t)}) \\
\leq & \left(\frac{1}{2\eta_\theta} + \frac{S_\omega}{2} + \frac{\mu}{2}\right) \left\|\theta^{(t+1)} - \theta^{(t)}\right\|_2^2 + \left(\frac{1}{2\eta_\theta} + \frac{\mu}{2}\right) \left\|\theta^{(t)} - \theta^{(t-1)}\right\|_2^2 \\
& + \frac{S_\omega}{2} \left\|\omega^{(t)} - \omega^{(t-1)}\right\|_2^2 - \langle\xi_\theta^{(t-1)}, \theta^{(t+1)} - \theta^{(t)}\rangle \\
\leq & \left(\frac{1}{2\eta_\theta} + \frac{S_\omega}{2} + \mu\right) \left\|\theta^{(t+1)} - \theta^{(t)}\right\|_2^2 + \left(\frac{1}{2\eta_\theta} + \frac{\mu}{2}\right) \left\|\theta^{(t)} - \theta^{(t-1)}\right\|_2^2 \\
& + \frac{S_\omega}{2} \left\|\omega^{(t)} - \omega^{(t-1)}\right\|_2^2 + \frac{\left\|\xi_\theta^{(t-1)}\right\|_2^2}{2\mu}.
\end{aligned}
\tag{39}
$$

Finally, taking expectation of (39) with respect to the noise and together with (27) , we prove the final result.

$$
\begin{aligned}
\mathbb{E}F(\omega^{(t+1)}, &\theta^{(t+1)}) - \mathbb{E}F(\omega^{(t)}, \theta^{(t)}) \\
\leq & \left(L_\omega - \frac{1}{\eta_\omega}\right) \mathbb{E}\|\omega^{(t+1)} - \omega^{(t)}\|_2^2 + \frac{S_\omega}{2}\mathbb{E}\left\|\omega^{(t)} - \omega^{(t-1)}\right\|_2^2 \\
& + \left(\frac{1}{2\eta_\theta} + \frac{S_\omega}{2} + \mu\right) \mathbb{E}\left\|\theta^{(t+1)} - \theta^{(t)}\right\|_2^2 \\
& + \left(\frac{1}{2\eta_\theta} + \frac{\mu}{2}\right) \mathbb{E}\left\|\theta^{(t)} - \theta^{(t-1)}\right\|_2^2 + \eta_\omega\mathbb{E}\left\|\xi_\omega^{(t)}\right\|_2^2 + \frac{1}{2\mu}\mathbb{E}\left\|\xi_\theta^{(t-1)}\right\|_2^2.
\end{aligned}
$$

$\square$

We then characterize the update of $\omega$.

**Lemma 7.** The update of $\omega$ satisfies

$$
\begin{aligned}
\mathbb{E}&\left\langle\omega^{(t+1)} - \omega^{(t)} - (\omega^{(t)} - \omega^{(t-1)}), \omega^{(t+1)} - \omega^{(t)}\right\rangle \\
\leq & -\frac{\eta_\omega}{\eta_\theta} \cdot \mathbb{E}\left\langle(\theta^{(t+2)} - \theta^{(t+1)}) - (\theta^{(t+1)} - \theta^{(t)}) - (\epsilon^{(t+2)} - \epsilon^{(t+1)}), \theta^{(t+1)} - \theta^{(t)}\right\rangle \\
& - \frac{\mu\eta_\omega}{2} \cdot \mathbb{E}\left\|\theta^{(t+1)} - \theta^{(t)}\right\|_2^2 + \frac{\eta_\omega(5L_\omega + 1)}{2} \cdot \mathbb{E}\left\|\omega^{(t+1)} - \omega^{(t)}\right\|_2^2 + \frac{\eta_\omega L_\omega}{2} \cdot \left\|\omega^{(t)} - \omega^{(t-1)}\right\|_2^2 \\
& + (\frac{\eta_\omega}{2\mu} + \eta_\omega^2)\mathbb{E}\left\|\xi_\omega^{(t)}\right\|_2^2 + \frac{\eta_\omega}{2}\mathbb{E}\left\|\xi_\omega^{(t-1)}\right\|_2^2.
\end{aligned}
$$

*Proof.* By the update policy, we have

$$
\mathbb{E} \left\langle \omega^{(t+1)} - \omega^{(t)} - (\omega^{(t)} - \omega^{(t-1)}), \omega^{(t+1)} - \omega^{(t)} \right\rangle
$$

$$
= -\eta_\omega \mathbb{E} \left\langle \nabla_\omega F(\omega^{(t)}, \theta^{(t+1)}) + \xi_\omega^{(t)} - \nabla_\omega F(\omega^{(t-1)}, \theta^{(t)}) - \xi_\omega^{(t-1)}, \omega^{(t+1)} - \omega^{(t)} \right\rangle
$$

$$
= -\eta_\omega \mathbb{E} \left\langle \nabla_\omega F(\omega^{(t)}, \theta^{(t+1)}) - \nabla_\omega F(\omega^{(t)}, \theta^{(t)}), \omega^{(t+1)} - \omega^{(t)} \right\rangle
$$

$$
\quad -\eta_\omega \mathbb{E} \left\langle \nabla_\omega F(\omega^{(t)}, \theta^{(t)}) - \nabla_\omega F(\omega^{(t-1)}, \theta^{(t)}), \omega^{(t+1)} - \omega^{(t)} \right\rangle
$$

$$
\quad -\eta_\omega \mathbb{E} \left\langle \xi_\omega^{(t)}, -\eta_\omega (\nabla_\omega F(\omega^{(t)}, \theta^{(t+1)}) + \xi_\omega^{(t)}) \right\rangle + \eta_\omega \mathbb{E} \left\langle \xi_\omega^{(t-1)}, \omega^{(t+1)} - \omega^{(t)} \right\rangle
$$

$$
\leq \underbrace{-\eta_\omega \mathbb{E} \left\langle \nabla_\omega F(\omega^{(t)}, \theta^{(t+1)}) - \nabla_\omega F(\omega^{(t)}, \theta^{(t)}), \omega^{(t+1)} - \omega^{(t)} \right\rangle}_{(C)}
$$

$$
\underbrace{-\eta_\omega \mathbb{E} \left\langle \nabla_\omega F(\omega^{(t)}, \theta^{(t)}) - \nabla_\omega F(\omega^{(t-1)}, \theta^{(t)}), \omega^{(t+1)} - \omega^{(t)} \right\rangle}_{(D)}
$$

$$
+ \eta_\omega^2 \mathbb{E} \left\| \xi_\omega^{(t)} \right\|_2^2 + \frac{\eta_\omega}{2} \mathbb{E} \left\| \xi_\omega^{(t-1)} \right\|_2^2 + \frac{\eta_\omega}{2} \mathbb{E} \left\| \omega^{(t+1)} - \omega^{(t)} \right\|_2^2. \tag{40}
$$

Then for $(C)$, by the definition of objective function, we have

$$
(C) = -\eta_\omega \mathbb{E} \left\langle \nabla_\omega \sum_j G(\widetilde{\pi}_{\omega^{(t)}})_j (\theta^{(t+1)} - \theta^{(t)})_j, \omega^{(t+1)} - \omega^{(t)} \right\rangle
$$

$$
= -\eta_\omega \mathbb{E} \sum_j \left\langle \nabla_\omega G(\widetilde{\pi}_{\omega^{(t)}})_j - \nabla_\omega G(\widetilde{\pi}_{\widetilde{\omega}_j^{(t)}})_j + \nabla_\omega G(\widetilde{\pi}_{\widetilde{\omega}_j^{(t)}})_j), \omega^{(t+1)} - \omega^{(t)} \right\rangle \cdot
$$

$$
(\theta^{(t+1)} - \theta^{(t)})_j
$$

$$
= -\eta_\omega \mathbb{E} \sum_j (\theta^{(t+1)} - \theta^{(t)})_j \left\langle \nabla_\omega G(\widetilde{\pi}_{\omega^{(t)}})_j - \nabla_\omega G(\widetilde{\pi}_{\widetilde{\omega}_j^{(t)}})_j), \omega^{(t+1)} - \omega^{(t)} \right\rangle
$$

$$
\quad -\eta_\omega \mathbb{E} \sum_j (\theta^{(t+1)} - \theta^{(t)})_j \left\langle \nabla_\omega G(\widetilde{\pi}_{\widetilde{\omega}_j^{(t)}})_j), \omega^{(t+1)} - \omega^{(t)} \right\rangle, \tag{41}
$$

where for every $j$, $\widetilde{\pi}_{\widetilde{\omega}_j^{(t)}}$ is some interpolation between vectors $\widetilde{\pi}_{\omega^{(t)}}$ and $\widetilde{\pi}_{\omega^{(t+1)}}$ such that

$$
\left\langle \nabla_\omega G(\widetilde{\pi}_{\widetilde{\omega}_j^{(t)}})_j, \omega^{(t+1)} - \omega^{(t)} \right\rangle = G(\widetilde{\pi}_{\omega^{(t+1)}})_j - G(\widetilde{\pi}_{\omega^{(t)}})_j. \tag{42}
$$

For the first term in the right hand side of (41), by Lemma 4, we have

$$
-\eta_\omega \mathbb{E} \sum_j (\theta^{(t+1)} - \theta^{(t)})_j \left\langle \nabla_\omega G(\widetilde{\pi}_{\omega^{(t)}})_j - \nabla_\omega G(\widetilde{\pi}_{\widetilde{\omega}_j^{(t)}})_j), \omega^{(t+1)} - \omega^{(t)} \right\rangle
$$

$$
\leq 2\eta_\omega L_\omega \mathbb{E} \left\| \omega^{(t+1)} - \omega^{(t)} \right\|_2^2. \tag{43}
$$

For the second term in the right hand side of (41), by (42) we have

$$
-\eta_\omega \mathbb{E} \sum_j (\theta^{(t+1)} - \theta^{(t)})_j \left\langle \nabla_\omega G(\widetilde{\pi}_{\widetilde{\omega}_j^{(t)}})_j, \omega^{(t+1)} - \omega^{(t)} \right\rangle
$$

$$
= -\eta_\omega \mathbb{E} \left\langle G(\widetilde{\pi}_{\omega^{(t+1)}}) - G(\widetilde{\pi}_{\omega^{(t)}}), \theta^{(t+1)} - \theta^{(t)} \right\rangle.
$$

Then by the defintion of objective function, we have

$$
- \eta_\omega \mathbb{E} \sum_j (\theta^{(t+1)} - \theta^{(t)})_j \left\langle \nabla_\omega G(\widetilde{\pi}_{\widetilde{\omega}_j^{(t)}})_j, \omega^{(t+1)} - \omega^{(t)} \right\rangle
$$

$$
= - \eta_\omega \mathbb{E} \Big\langle \frac{1}{\eta_\theta} \left( \theta^{(t+2)} - \theta^{(t+1)} - \epsilon^{(t+2)} \right) - \xi_\theta^{(t+1)} + \mu \theta^{(t+1)}
$$

$$
- \frac{1}{\eta_\theta} \left( \theta^{(t+1)} - \theta^{(t)} - \epsilon^{(t+1)} \right) + \xi_\theta^{(t)} - \mu \theta^{(t)}, \theta^{(t+1)} - \theta^{(t)} \Big\rangle
$$

$$
= - \frac{\eta_\omega}{\eta_\theta} \mathbb{E} \left\langle (\theta^{(t+2)} - \theta^{(t+1)}) - (\theta^{(t+1)} - \theta^{(t)}) - (\epsilon^{(t+2)} - \epsilon^{(t+1)}), \theta^{(t+1)} - \theta^{(t)} \right\rangle
$$

$$
+ \eta_\omega \mathbb{E} \langle \xi_\theta^{(t+1)} - \xi_\theta^{(t)}, \theta^{(t+1)} - \theta^{(t)} \rangle - \mu \eta_\omega \mathbb{E} \left\| \theta^{(t+1)} - \theta^{(t)} \right\|_2^2
$$

$$
\leq - \frac{\eta_\omega}{\eta_\theta} \mathbb{E} \left\langle (\theta^{(t+2)} - \theta^{(t+1)}) - (\theta^{(t+1)} - \theta^{(t)}) - (\epsilon^{(t+2)} - \epsilon^{(t+1)}), \theta^{(t+1)} - \theta^{(t)} \right\rangle
$$

$$
- \frac{\mu \eta_\omega}{2} \mathbb{E} \left\| \theta^{(t+1)} - \theta^{(t)} \right\|_2^2 + \frac{\eta_\omega}{2\mu} \mathbb{E} \left\| \xi_\theta^{(t)} \right\|_2^2. \tag{44}
$$

For $(D)$, applying Cauchy-Schwartz inequality we have

$$
- \eta_\omega \mathbb{E} \langle \nabla_\omega F(\omega^{(t)}, \theta^{(t)}) - \nabla_\omega F(\omega^{(t-1)}, \theta^{(t)}), \omega^{(t+1)} - \omega^{(t)} \rangle
$$

$$
\leq \eta_\omega L_\omega \mathbb{E} \left\| \omega^{(t+1)} - \omega^{(t)} \right\|_2 \left\| \omega^{(t)} - \omega^{(t-1)} \right\|_2
$$

$$
\leq \frac{\eta_\omega L_\omega}{2} \mathbb{E} \left\| \omega^{(t+1)} - \omega^{(t)} \right\|_2^2 + \frac{\eta_\omega L_\omega}{2} \mathbb{E} \left\| \omega^{(t)} - \omega^{(t-1)} \right\|_2^2. \tag{45}
$$

Finally, combining (40)-(45), we prove Lemma 7. $\qquad \square$

For notational simplicity, we define

$$
\delta^{(t+2)} = (\theta^{(t+2)} - \theta^{(t+1)}) - (\theta^{(t+1)} - \theta^{(t)}). \tag{46}
$$

**Lemma 8.** The first term on the right hand side of Lemma (7) satisfies

$$
- \frac{\eta_\omega}{\eta_\theta} \mathbb{E} \left\langle (\theta^{(t+2)} - \theta^{(t+1)}) - (\theta^{(t+1)} - \theta^{(t)}) - (\epsilon^{(t+2)} - \epsilon^{(t+1)}), \theta^{(t+1)} - \theta^{(t)} \right\rangle \tag{47}
$$

$$
\leq - \frac{\eta_\omega}{2\eta_\theta} \mathbb{E} \left\| \theta^{(t+2)} - \theta^{(t+1)} \right\|_2^2 + \left( \frac{3\mu^2 \eta_\omega \eta_\theta}{2} + \frac{\eta_\omega}{2\eta_\theta} \right) \mathbb{E} \left\| \theta^{(t+1)} - \theta^{(t)} \right\|_2^2
$$

$$
+ \frac{3\eta_\omega \eta_\theta S_\omega^2}{2} \cdot \mathbb{E} \left\| \omega^{(t+1)} - \omega^{(t)} \right\|_2^2 + \frac{3}{2} \eta_\omega \eta_\theta \left( \mathbb{E} \left\| \xi_\theta^{(t+1)} \right\|_2^2 + \mathbb{E} \left\| \xi_\theta^{(t)} \right\|_2^2 \right). \tag{48}
$$

*Proof.* Plugging (46) into (47), we obtain

$$
- \frac{\eta_\omega}{\eta_\theta} \mathbb{E} \left\langle (\theta^{(t+2)} - \theta^{(t+1)}) - (\theta^{(t+1)} - \theta^{(t)}) - (\epsilon^{(t+2)} - \epsilon^{(t+1)}), \theta^{(t+1)} - \theta^{(t)} \right\rangle
$$

$$
= \frac{\eta_\omega}{\eta_\theta} \mathbb{E} \left\langle \delta^{(t+2)} - (\epsilon^{(t+2)} - \epsilon^{(t+1)}), \delta^{(t+2)} - (\theta^{(t+2)} - \theta^{(t+1)}) \right\rangle
$$

$$
= \frac{\eta_\omega}{\eta_\theta} \mathbb{E} \left\langle \delta^{(t+2)} - (\epsilon^{(t+2)} - \epsilon^{(t+1)}), \delta^{(t+2)} \right\rangle - \frac{\eta_\omega}{\eta_\theta} \mathbb{E} \left\langle \delta^{(t+2)}, \theta^{(t+2)} - \theta^{(t+1)} \right\rangle
$$

$$
+ \frac{\eta_\omega}{\eta_\theta} \mathbb{E} \left\langle (\epsilon^{(t+2)} - \epsilon^{(t+1)}), \theta^{(t+2)} - \theta^{(t+1)} \right\rangle. \tag{49}
$$

By applying the equality

$$
\langle u, v \rangle = \frac{1}{2} (\|u\|_2^2 + \|v\|_2^2 - \|u - v\|_2^2)
$$

to the first two terms on the right hand side of (49), we obtain

$$
-\frac{\eta_\omega}{\eta_\theta}\mathbb{E}\left\langle (\theta^{(t+2)}-\theta^{(t+1)})-(\theta^{(t+1)}-\theta^{(t)})-(\epsilon^{(t+2)}-\epsilon^{(t+1)}),\theta^{(t+1)}-\theta^{(t)}\right\rangle
$$

$$
=\frac{\eta_\omega}{2\eta_\theta}\mathbb{E}\left(\left\|\delta^{(t+2)}-(\epsilon^{(t+2)}-\epsilon^{(t+1)})\right\|_2^2+\left\|\delta^{(t+2)}\right\|_2^2-\left\|\epsilon^{(t+2)}-\epsilon^{(t+1)}\right\|_2^2\right)
$$

$$
-\frac{\eta_\omega}{2\eta_\theta}\mathbb{E}\left(\left\|\delta^{(t+2)}\right\|_2^2+\left\|\theta^{(t+2)}-\theta^{(t+1)}\right\|_2^2-\left\|\theta^{(t+1)}-\theta^{(t)}\right\|_2^2\right)
$$

$$
+\frac{\eta_\omega}{\eta_\theta}\mathbb{E}\langle\epsilon^{(t+2)}-\epsilon^{(t+1)},\theta^{(t+2)}-\theta^{(t+1)}\rangle. \tag{50}
$$

Recall that

$$
\theta^{(t+2)}=\Pi_\kappa\left(\theta^{(t+1)}+\eta_\theta\left(\nabla_\theta F(\omega^{(t+1)}\theta^{(t+1)})+\xi_\theta^{(t+1)}\right)\right),
$$

and

$$
\theta^{(t+1)}=\Pi_\kappa\left(\theta^{(t)}+\eta_\theta\left(\nabla_\theta F(\omega^{(t)},\theta^{(t)})+\xi_\theta^{(t)}\right)\right).
$$

Following from the convexity of $\{\theta|\,\|\theta\|_2\leq\kappa\}$, we have

$$
\langle\epsilon^{(t+2)},\theta^{(t+2)}-\theta^{(t+1)}\rangle\leq 0\quad\text{and}\quad\langle\epsilon^{(t+1)},\theta^{(t+2)}-\theta^{(t+1)}\rangle\geq 0.
$$

Thus, the last term on the right side of (50) is negative. By rearranging the terms in (50), we obtain

$$
-\frac{\eta_\omega}{\eta_\theta}\mathbb{E}\left\langle(\theta^{(t+2)}-\theta^{(t+1)})-(\theta^{(t+1)}-\theta^{(t)})-(\epsilon^{(t+2)}-\epsilon^{(t+1)}),\theta^{(t+1)}-\theta^{(t)}\right\rangle
$$

$$
\leq\frac{\eta_\omega}{2\eta_\theta}\cdot\mathbb{E}\left\|\delta^{(t+2)}-(\epsilon^{(t+2)}-\epsilon^{(t+1)})\right\|_2^2-\frac{\eta_\omega}{2\eta_\theta}\mathbb{E}\left\|\theta^{(t+2)}-\theta^{(t+1)}\right\|_2^2+\frac{\eta_\omega}{2\eta_\theta}\mathbb{E}\left\|\theta^{(t+1)}-\theta^{(t)}\right\|_2^2. \tag{51}
$$

By definition of $\delta^{(t+2)}$ in (46), we have

$$
\delta^{(t+2)}-(\epsilon^{(t+2)}-\epsilon^{(t+1)})=(\theta^{(t+2)}-\theta^{(t+1)}-\epsilon^{(t+2)})-(\theta^{(t+1)}-\theta^{(t)}-\epsilon^{(t+1)})
$$

$$
=\eta_\theta[G(\widetilde{\pi}_{\omega^{(t+1)}})-G(\widetilde{\pi}_{\omega^{(t)}})-\mu(\theta^{(t+1)}-\theta^{(t)})+\xi_\theta^{(t+1)}-\xi_\theta^{(t)}].
$$

Using the Cauchy-Schwarz inequality, we obtain

$$
\frac{\eta_\omega}{2\eta_\theta}\cdot\mathbb{E}\left\|\delta^{(t+2)}-(\epsilon^{(t+2)}-\epsilon^{(t+1)})\right\|_2^2
$$

$$
\leq\frac{3\eta_\omega\eta_\theta}{2}\cdot\left(\mathbb{E}\left\|G(\widetilde{\pi}_{\omega^{(t+1)}})-G(\widetilde{\pi}_{\omega^{(t)}})\right\|_2^2+\mathbb{E}\left\|\mu(\theta^{(t+1)}-\theta^{(t)})\right\|_2^2+\mathbb{E}\left\|\xi_\theta^{(t+1)}-\xi_\theta^{(t)}\right\|_2^2\right)
$$

$$
\leq\frac{3\eta_\omega\eta_\theta S_\omega^2}{2}\cdot\mathbb{E}\left\|\omega^{(t+1)}-\omega^{(t)}\right\|_2^2+\frac{3}{2}\mu^2\eta_\omega\eta_\theta\cdot\mathbb{E}\left\|\theta^{(t+1)}-\theta^{(t)}\right\|_2^2+\frac{3\eta_\omega\eta_\theta}{2}\cdot\mathbb{E}\left\|\xi_\theta^{(t+1)}-\xi_\theta^{(t)}\right\|_2^2. \tag{52}
$$

Plugging (52) into (51) yields (48), which concludes the proof of Lemma 8. $\qquad\square$

**Lemma 9.** For the update of $\omega$, we have

$$
\frac{1}{2}\cdot\mathbb{E}\left\|\omega^{(t+1)}-\omega^{(t)}\right\|_2^2-\frac{1}{2}\cdot\mathbb{E}\left\|\omega^{(t)}-\omega^{(t-1)}\right\|_2^2
$$

$$
\leq\left(\frac{\eta_\omega(5L_\omega+1)}{2}+\frac{3\eta_\omega\eta_\theta S_\omega^2}{2}\right)\mathbb{E}\left\|\omega^{(t+1)}-\omega^{(t)}\right\|_2^2+\frac{\eta_\omega L_\omega}{2}\cdot\mathbb{E}\left\|\omega^{(t)}-\omega^{(t-1)}\right\|_2^2
$$

$$
-\frac{\eta_\omega}{2\eta_\theta}\cdot\mathbb{E}\left\|\theta^{(t+2)}-\theta^{(t+1)}\right\|_2^2+\left(\frac{3}{2}\eta_\omega\eta_\theta\mu^2+\frac{\eta_\omega}{2\eta_\theta}-\frac{\mu\eta_\omega}{2}\right)\mathbb{E}\left\|\theta^{(t+1)}-\theta^{(t)}\right\|_2^2
$$

$$
+(\eta_\omega^2+\frac{\eta_\omega}{2\mu})\mathbb{E}\left\|\xi_\omega^{(t)}\right\|_2^2+\frac{\eta_\omega}{2}\cdot\mathbb{E}\left\|\xi_\omega^{(t-1)}\right\|_2^2+\frac{3}{2}\eta_\omega\eta_\theta\cdot\left(\mathbb{E}\left\|\xi_\theta^{(t+1)}\right\|_2^2+\mathbb{E}\left\|\xi_\theta^{(t)}\right\|_2^2\right).
$$

*Proof.* Combing Lemma 7 and Lemma 8, we obtain

$$\mathbb{E}\langle \omega^{(t+1)} - \omega^{(t)} - (\omega^{(t)} - \omega^{(t-1)}), \omega^{(t+1)} - \omega^{(t)}\rangle$$

$$\leq -\frac{\eta_\omega}{2\eta_\theta}\mathbb{E}\left\|\theta^{(t+2)} - \theta^{(t+1)}\right\|_2^2 + \left(\frac{3}{2}\eta_\omega\eta_\theta\mu^2 + \frac{\eta_\omega}{2\eta_\theta}\right)\mathbb{E}\left\|\theta^{(t+1)} - \theta^{(t)}\right\|_2^2$$

$$+ \frac{3\eta_\omega\eta_\theta S_\omega^2}{2}\mathbb{E}\left\|\omega^{(t+1)} - \omega^{(t)}\right\|_2^2 - \frac{\mu\eta_\omega}{2}\mathbb{E}\left\|\theta^{(t+1)} - \theta^{(t)}\right\|_2^2$$

$$+ 2\eta_\omega L_\omega\mathbb{E}\left\|\omega^{(t+1)} - \omega^{(t)}\right\|_2^2 + \frac{\eta_\omega(L_\omega + 1)}{2}\mathbb{E}\left\|\omega^{(t+1)} - \omega^{(t)}\right\|_2^2 + \frac{\eta_\omega L_\omega}{2}\mathbb{E}\left\|\omega^{(t)} - \omega^{(t-1)}\right\|_2^2$$

$$+ (\eta_\omega^2 + \frac{\eta_\omega}{2\mu})\mathbb{E}\left\|\xi_\omega^{(t)}\right\|_2^2 + \frac{\eta_\omega}{2}\mathbb{E}\left\|\xi_\omega^{(t-1)}\right\|_2^2 + \frac{3}{2}\eta_\omega\eta_\theta\cdot(\mathbb{E}\left\|\xi_\theta^{(t+1)}\right\|_2^2 + \mathbb{E}\left\|\xi_\theta^{(t)}\right\|_2^2). \tag{53}$$

Note that we have

$$\mathbb{E}\langle \omega^{(t+1)} - \omega^{(t)} - (\omega^{(t)} - \omega^{(t-1)}), \omega^{(t+1)} - \omega^{(t)}\rangle$$

$$= \frac{1}{2}\mathbb{E}\left\|\omega^{(t+1)} - \omega^{(t)}\right\|_2^2 - \frac{1}{2}\mathbb{E}\left\|\omega^{(t)} - \omega^{(t-1)}\right\|_2^2 + \frac{1}{2}\mathbb{E}\left\|(\omega^{(t+1)} - \omega^{(t)}) - (\omega^{(t)} - \omega^{(t-1)})\right\|_2^2$$

$$\geq \frac{1}{2}\mathbb{E}\left\|\omega^{(t+1)} - \omega^{(t)}\right\|_2^2 - \frac{1}{2}\mathbb{E}\left\|\omega^{(t)} - \omega^{(t-1)}\right\|_2^2. \tag{54}$$

Combining (53) and (54), we conclude the proof of Lemma 9. $\qquad\square$

**Lemma 10.** Suppose Assumption 1, 3 and 5 hold. $F(\omega^{(t)}, \theta^{(t)})$ is lower bounded throughout all iterations.

*Proof.* By definition of $F$ in (2), we have

$$F(\omega^{(t)}, \theta^{(t)}) \geq \mathbb{E}_{\widetilde{\pi}_{\omega^{(t)}}}\widetilde{r}_{\theta^{(t)}}(s, a) - \mathbb{E}_{\pi^*}\widetilde{r}_{\theta^{(t)}}(\psi_s, \psi_a) - \frac{\mu}{2}\|\theta\|_2^2 - \lambda B_H$$

$$\geq -\left(2\sqrt{2}\rho_g\kappa + \frac{\mu}{2}\kappa^2 + \lambda B_H\right). \tag{55}$$

$\qquad\square$

### B.4 PROOF OF LEMMA 1

Recall that we construct a potential function that dacays monotonically along the solution path, which takes the form

$$\mathcal{E}^{(t+1)} = \mathbb{E}F(\omega^{(t+1)}, \theta^{(t+1)}) + s\cdot\left(\frac{1 + 2\eta_\omega L_\omega}{2}\mathbb{E}\left\|\omega^{(t+1)} - \omega^{(t)}\right\|_2^2\right.$$

$$\left. + \left(\frac{\eta_\omega}{2\eta_\theta} - \frac{\mu\eta_\omega}{4} + \frac{3}{2}\eta_\omega\eta_\theta\mu^2\right)\mathbb{E}\left\|\theta^{(t+2)} - \theta^{(t+1)}\right\|_2^2 + \frac{\mu\eta_\omega}{8}\mathbb{E}\left\|\theta^{(t+1)} - \theta^{(t)}\right\|_2^2\right). \tag{56}$$

for some constant $s > 0$. We define five constants $k_1, k_2, k_3, k_4, k_5$ as

$$k_1 = \frac{1}{2\eta_\omega} - s\cdot\left(\frac{\eta_\omega(7L_\omega + 1)}{2} + \frac{3\eta_\omega\eta_\theta S_\omega^2}{2}\right), \quad k_2 = s\cdot\frac{\eta_\omega L_\omega}{2} - \frac{S_\omega}{2},$$

$$k_3 = s\cdot\left(\frac{\eta_\omega\mu}{4} - \frac{3}{2}\eta_\omega\eta_\theta\mu^2\right), \qquad k_5 = s\cdot\frac{\mu\eta_\omega}{8} - \left(\frac{1}{2\eta_\theta} + \frac{\mu}{2}\right),$$

$$k_4 = s\cdot\frac{\mu\eta_\omega}{8} - \left(\frac{1}{2\eta_\theta} + \frac{S_\omega + 2\mu}{2}\right).$$

Here, we restate Lemma 1 and then prove it.

**Lemma 1.** We choose step sizes $\eta_\theta, \eta_\omega$ satisfying

$$\eta_\omega \leq \min\left\{\frac{L_\omega}{S_\omega(8L_\omega + 2)}, \frac{1}{2L_\omega}\right\}, \quad \eta_\theta \leq \min\left\{\frac{1}{150\mu}, \frac{7L_\omega + 1}{150S_\omega^2}, \frac{1}{100(2\mu + S_\omega)}\right\},$$

and meanwhile $\eta_\omega/\eta_\theta \leq \mu/(30L_\omega + 5)$, where $L_\omega = \sqrt{2}\kappa\rho_g S_\pi |\mathcal{A}| + \lambda S_H$. Then we have

$$
\begin{aligned}
\mathcal{E}^{(t+1)} - \mathcal{E}^{(t)} \leq &- k_1 \mathbb{E} \left\| \omega^{(t+1)} - \omega^{(t)} \right\|_2^2 - k_2 \mathbb{E} \left\| \omega^{(t)} - \omega^{(t-1)} \right\|_2^2 - k_3 \mathbb{E} \left\| \theta^{(t+2)} - \theta^{(t+1)} \right\|_2^2 \\
&- k_4 \mathbb{E} \left\| \theta^{(t+1)} - \theta^{(t)} \right\|_2^2 - k_5 \mathbb{E} \left\| \theta^{(t)} - \theta^{(t-1)} \right\|_2^2 \\
&+ \eta_\omega \mathbb{E} \left\| \xi_\omega^{(t)} \right\|_2^2 + \frac{1}{2} \mathbb{E} \left\| \xi_\theta^{(t-1)} \right\|_2^2 \\
&+ s \cdot \left( \eta_\omega^2 \mathbb{E} \left\| \xi_\omega^{(t)} \right\|_2^2 + \frac{\eta_\omega}{2} \mathbb{E} \left\| \xi_\omega^{(t-1)} \right\|_2^2 + \frac{3\eta_\omega\eta_\theta}{2} \cdot (\mathbb{E} \left\| \xi_\theta^{(t+1)} \right\|_2^2 + \mathbb{E} \left\| \xi_\theta^{(t)} \right\|_2^2) \right). \quad (57)
\end{aligned}
$$

Moreover, we have $k_1, k_2, k_3, k_4, k_5 > 0$ for

$$
s = \frac{8}{\eta_\omega^2(58L_\omega + 9)}.
$$

*Proof.* For notational simplicity, we define

$$
\begin{aligned}
K^{(t+1)} = &\frac{1 + 2\eta_\omega L_\omega}{2} \mathbb{E} \left\| \omega^{(t+1)} - \omega^{(t)} \right\|_2^2 \\
&+ \left( \frac{\eta_\omega}{2\eta_\theta} - \frac{\mu\eta_\omega}{4} + \frac{3\eta_\omega\eta_\theta\mu^2}{2} \right) \cdot \mathbb{E} \left\| \theta^{(t+2)} - \theta^{(t+1)} \right\|_2^2 + \frac{\mu\eta_\omega}{8} \mathbb{E} \left\| \theta^{(t+1)} - \theta^{(t)} \right\|_2^2. \quad (58)
\end{aligned}
$$

By rearranging the inequality in Lemma (9), we obtain

$$
\begin{aligned}
K^{(t+1)} - K^{(t)} \leq &\left( \frac{\eta_\omega(7L_\omega + 1)}{2} + \frac{3\eta_\omega\eta_\theta S_\omega^2}{2} \right) \mathbb{E} \left\| \omega^{(t+1)} - \omega^{(t)} \right\|_2^2 \\
&- \frac{\eta_\omega L_\omega}{2} \mathbb{E} \left\| \omega^{(t)} - \omega^{(t-1)} \right\|_2^2 - \left( \frac{\mu\eta_\omega}{4} - \frac{3\mu^2\eta_\omega\eta_\theta}{2} \right) \mathbb{E} \left\| \theta^{(t+2)} - \theta^{(t+1)} \right\|_2^2 \\
&- \frac{\mu\eta_\omega}{8} \mathbb{E} \left\| \theta^{(t+1)} - \theta^{(t)} \right\|_2^2 - \frac{\mu\eta_\omega}{8} \mathbb{E} \left\| \theta^{(t)} - \theta^{(t-1)} \right\|_2^2 \\
&+ (\eta_\omega^2 + \frac{\eta_\omega}{2\mu}) \mathbb{E} \left\| \xi_\omega^{(t)} \right\|_2^2 + \frac{\eta_\omega}{2} \mathbb{E} \left\| \xi_\omega^{(t-1)} \right\|_2^2 + \frac{3\eta_\omega\eta_\theta}{2} \cdot (\mathbb{E} \left\| \xi_\theta^{(t+1)} \right\|_2^2 + \mathbb{E} \left\| \xi_\theta^{(t)} \right\|_2^2). \quad (59)
\end{aligned}
$$

By definition of $P^{(t)}$ in (56), we have

$$
\mathcal{E}^{(t)} = F(\omega^{(t)}, \theta^{(t)}) + s \cdot K^{(t)}
$$

for some constant $s > 0$. Combining (59) and Lemma 6, since $\eta_\omega < \frac{1}{2L_\omega}$, we obtain

$$
\begin{aligned}
\mathcal{E}^{(t+1)} - \mathcal{E}^{(t)} \leq &- \left( \frac{1}{2\eta_\omega} - s \cdot \left( \frac{\eta_\omega(7L_\omega + 1)}{2} + \frac{3\eta_\omega\eta_\theta S_\omega^2}{2} \right) \right) \left\| \omega^{(t+1)} - \omega^{(t)} \right\|_2^2 \\
&- \left( s \cdot \frac{\eta_\omega L_\omega}{2} - \frac{S_\omega}{2} \right) \left\| \omega^{(t)} - \omega^{(t-1)} \right\|_2^2 - s \cdot \left( \frac{\mu\eta_\omega}{4} - \frac{3\mu^2\eta_\omega\eta_\theta}{2} \right) \left\| \theta^{(t+2)} - \theta^{(t+1)} \right\|_2^2 \\
&- \left( s \cdot \frac{\mu\eta_\omega}{8} - \left( \frac{1}{2\eta_\theta} + \frac{2\mu + S_\omega}{2} \right) \right) \left\| \theta^{(t+1)} - \theta^{(t)} \right\|_2^2 \\
&- \left( s \cdot \frac{\mu\eta_\omega}{8} - \left( \frac{1}{2\eta_\theta} + \frac{\mu}{2} \right) \right) \left\| \theta^{(t)} - \theta^{(t-1)} \right\|_2^2 \\
&+ \eta_\omega \mathbb{E} \left\| \xi_\omega^{(t)} \right\|_2^2 + \frac{1}{2\mu} \mathbb{E} \left\| \xi_\theta^{(t-1)} \right\|_2^2 \\
&+ s \cdot \left( (\eta_\omega^2 + \frac{\eta_\omega}{2\mu}) \mathbb{E} \left\| \xi_\omega^{(t)} \right\|_2^2 + \frac{\eta_\omega}{2} \mathbb{E} \left\| \xi_\omega^{(t-1)} \right\|_2^2 + \frac{3\eta_\omega\eta_\theta}{2} \cdot (\mathbb{E} \left\| \xi_\theta^{(t+1)} \right\|_2^2 + \mathbb{E} \left\| \xi_\theta^{(t)} \right\|_2^2) \right). \\
&\qquad\qquad\qquad\qquad\qquad\qquad\qquad\qquad\qquad\qquad\qquad\qquad\qquad\qquad\qquad (60)
\end{aligned}
$$

Since $\eta_\theta < \frac{1}{150\mu}$, we have $k_3 > 0$. Now we choose a proper constant $s$ such that $k_1, k_2, k_4, k_5$ are positive. Note that they are positive if and only if

$$
1/(2\eta_\omega) > s \cdot ((\eta_\omega(7L_\omega + 1)/2 + \frac{3\eta_\omega\eta_\theta S_\omega^2}{2})), \quad (61)
$$

$$
s \cdot \eta_\omega L_\omega/2 > S_\omega/2, \quad (62)
$$

$$
s \cdot (\mu\eta_\omega/8) > 1/(2\eta_\theta) + (2\mu + S_\omega)/2, \quad (63)
$$

$$
s \cdot (\mu\eta_\omega)/8 > 1/(2\eta_\theta) + \mu/2. \quad (64)
$$

Rearranging the terms in (61), (62), (63), (64) , we obtain

$$\frac{S_\omega}{\eta_\omega L_\omega} < s < \frac{1/(2\eta_\omega)}{\eta_\omega(7L_\omega + 1 + 3\eta_\theta S_\omega^2)/2} \quad, \tag{65}$$

$$\frac{1/(2\eta_\theta) + (S_\omega + 2\mu)/2}{\mu\eta_\omega/8} < s < \frac{1/(2\eta_\omega)}{\eta_\omega(7L_\omega + 1 + 3\eta_\theta S_\omega^2)/2} \quad. \tag{66}$$

Since

$$\eta_\theta \leq \frac{7L_\omega + 1}{150 S_\omega^2} \text{ and } \eta_\theta < \frac{1}{100(2\mu + S_\omega)},$$

by rearranging the terms in (65) and (66) and taking the leading terms, we obtain

$$\eta_\omega < \frac{L_\omega}{S_\omega(8L_\omega + 2)} \text{ and } \frac{\eta_\omega}{\eta_\theta} < \frac{\mu}{30L_\omega + 5}.$$

Therefore, we have

$$\frac{\eta_\omega}{\eta_\theta} < \frac{\mu}{30L_\omega + 5}, \ \eta_\omega < \overline{\eta}_\omega \text{ and } \eta_\theta < \overline{\eta}_\theta,$$

where

$$\overline{\eta}_\omega = \min\left\{\frac{L_\omega}{S_\omega(8L_\omega + 2)}, \frac{1}{2L_\omega}\right\},$$

$$\overline{\eta}_\theta = \min\left\{\frac{1}{150\mu}, \frac{7L_\omega + 1}{150 S_\omega^2}, \frac{1}{100(2\mu + S_\omega)}\right\}.$$

□

### B.5 PROOF OF THEOREM 2

Let $k = 1/\min\{k_1, k_4\}$, and $\phi = \max\{1, 1/\eta_\omega^2, 1/\eta_\theta^2\}$. Then we have

$$NJ_N \leq \sum_{t=1}^{N} \phi \cdot \mathbb{E}\left(\left\|\omega^{(t+1)} - \omega^{(t)}\right\|_2^2 + \left\|\theta^{(t+1)} - \theta^{(t)}\right\|_2^2\right)$$

$$\leq \phi k \Big(\sum_{t=1}^{N}(\mathcal{E}^{(t)} - \mathcal{E}^{(t+1)}) + \sum_{t=1}^{N}\big((\eta_\omega + s\eta_\omega^2 + \frac{s\eta_\omega}{2\mu})\mathbb{E}\left\|\xi_\omega^{(t)}\right\|_2^2 + \frac{1}{2\mu}\cdot\mathbb{E}\left\|\xi_\theta^{(t-1)}\right\|_2^2$$

$$+ s\cdot(\eta_\omega/2\cdot\mathbb{E}\left\|\xi_\omega^{(t-1)}\right\|_2^2 + 3\eta_\omega\eta_\theta/2\cdot(\mathbb{E}\left\|\xi_\theta^{(t+1)}\right\|_2^2 + \mathbb{E}\left\|\xi_\theta^{(t)}\right\|_2^2))\Big)$$

$$\leq \phi k\Big((\mathcal{E}^{(1)} - \mathcal{E}^{(N)}) + \sum_{t=1}^{N}\big(2\max\left\{\eta_\omega + s\eta_\omega^2 + \frac{s\eta_\omega}{2\mu}, \frac{s\eta_\omega}{2}\right\}\mathbb{E}\left\|\xi_\omega^{(t)}\right\|_2^2$$

$$+ 3\max\left\{\frac{1}{2\mu}, \frac{3\eta_\omega\eta_\theta}{2}\right\}\mathbb{E}\left\|\xi_\theta^{(t)}\right\|_2^2)\Big). \tag{67}$$

Now set

$$\nu = \max\left\{2\max\left\{\eta_\omega + s\eta_\omega^2 + \frac{s\eta_\omega}{2\mu}, \frac{s\eta_\omega}{2}\right\}, 3\max\left\{\frac{1}{2\mu}, \frac{3\eta_\omega\eta_\theta}{2}\right\}\right\}$$

and divide both sides of (67) by $N$, we have

$$J_N \leq \frac{k\phi(\mathcal{E}^{(1)} - \mathcal{E}^{(N)})}{N} + k\phi\nu\left(\frac{M_G}{q_\omega} + \frac{M_\theta}{q_\theta}\right). \tag{68}$$

By definition of $\mathcal{E}$ in (56) and Lemma 10, we have

$$\mathcal{E}^{(N)} \geq F(\omega^{(N)}, \theta^{(N)}) \geq -\left(2\sqrt{2}\rho_g\kappa + \frac{\mu}{2}\kappa^2 + \lambda B_H\right) > -\infty.$$

Now for any given $\epsilon > 0$, we take

$$q_\theta = \frac{4k\phi\nu M_\theta}{\epsilon}, \ q_\omega = \frac{4k\phi\nu M_\omega}{\epsilon} \text{ and } N = k\phi\frac{2\mathcal{E}^{(1)} + 4\sqrt{2}\rho_g\kappa + \mu\kappa^2 + 2\lambda B_H}{\epsilon},$$

and obtain

$$J_N \leq \epsilon.$$

## C  EMPIRICAL MAXIMIZER CASE

Notice that the object function (2) is quadratic in $\theta$, and thus we are able to get the empirical maximizer by simple calculation instead of performing stochastic gradient ascent at each iteration. This allows us to avoid using large batches while achieving the same sample complexity $\widetilde{O}(\frac{1}{\epsilon^2})$.

Given a fixed $\omega^{(t)}$, by definition of G in (25), we can get the optimal $\theta^*(\omega^{(t)})$ in population form:

$$\theta^*(\omega^{(t)}) = \frac{1}{\mu}[G(\pi_{\omega^{(t)}}) - G(\pi^*)]. \tag{69}$$

At the $(t+1)$-th iteration, we can achieve the closed form of the empirical maximizer by sampling one trajectory randomly and then apply stochastic gradient descent to the outer minimization. More specifically, the updating rule is defined as

$$\widetilde{\theta}^{(t+1)}(\omega^{(t)}) = \frac{1}{\mu}[\widetilde{G}(\pi_{\omega^{(t)}}) - \widetilde{G}(\pi^*)], \tag{70}$$

$$\omega^{(t+1)} = \omega^{(t)} - \eta_\omega \nabla_\omega \widetilde{f}_t(\omega^{(t)}, \widetilde{\theta}^{(t+1)}), \tag{71}$$

where $\widetilde{G}$ is the empirical version of $G$ obtained by sampling, and $\nabla_\omega \widetilde{f}$ is the stochastic approximation of $\nabla_\omega F$ at the $(t+1)$-the iteration.

We define the stationary point as follow.

**Definition 4.** We call $\omega^*$ an stationary point if $\nabla_\omega F(\omega^*, \theta^*(\omega^*)) = 0$.

Before we proceed with the convergence analysis, we impose the following assumptions on the problem.

**Assumption 6.** There is some constant $M_G > 0$ s.t. for any $\omega, \widetilde{\theta}$ and $\pi$, the following two conditions hold.

$$\text{Unbiased}: \ \mathbb{E}\nabla_\omega \widetilde{f}_t(\omega, \widetilde{\theta}) = \nabla_\omega F(\omega, \widetilde{\theta}), \mathbb{E}\widetilde{G}(\pi) = \mathbb{E}G(\pi).$$

$$\text{Gradient bounded}: \ \mathbb{E}\left\|\nabla_\omega \widetilde{f}_t(\omega, \theta)\right\|_2^2 \leq M_G.$$

Assumption 6 requires the stochastic gradient to be unbiased with bounded second moment.

**Corollary 4.** Under Assumption 1, 3 and 5, there exists $B_F = \frac{12\rho_g^2}{\mu} + \lambda B_H$ such that for any $\omega$, we have $|F(\omega, \theta^*(\omega))| < B_F$.

*Proof.* By Equation (69),we have $\|\theta^*(\omega)\|_2 \leq \frac{2\sqrt{2}\rho_g}{\mu}$. Plugging this into $F(\omega, \theta^*(\omega))$, we have for any $\omega$,

$$|F(\omega, \theta^*(\omega))| = |\mathbb{E}_{\widetilde{\pi}_\omega} \langle \theta^*(\omega), g(\psi_{s_t}, \psi_{a_t}) \rangle - \mathbb{E}_{\pi^*} \langle \theta^*(\omega), g(\psi_{s_t}, \psi_{a_t}) \rangle$$
$$- \lambda H(\widetilde{\pi}_{\omega^{(t)}}) - \frac{\mu}{2} \|\theta^*(\omega)\|_2^2|$$
$$\leq 2 \|\theta^*(\omega)\|_2 \cdot \max_{s,a} \|g(\psi_s, \psi_a)\|_2 + \lambda H(\widetilde{\pi}_{\omega^{(t)}}) + \frac{\mu}{2} \|\theta^*(\omega)\|_2^2$$
$$\leq \frac{12\rho_g^2}{\mu} + \lambda M_H.$$

$\square$

For notational simplicity, We define $\xi_\omega^{(t)} = \nabla_\theta F(\omega^{(t)}, \widetilde{\theta}^{(t+1)}) - \nabla_\theta \widetilde{f}_t(\omega^{(t)}, \widetilde{\theta}^{(t+1)})$ as the i.i.d stochastic noise and $\sigma_\theta^{(t)} = \theta^*(\omega^{(t)}) - \widetilde{\theta}^{(t)}$ as the i.i.d error of empirical maximizer. We denote $\mathbb{E}$ as the expectation over $\xi_\omega^{(t)}(t \geq 1)$ and $\sigma_\theta^{(t)}(t \geq 1)$. We measure the sub-stationarity of the algotithm at the iteration $N$ by

$$I_N = \min_{1 \leq t \leq N} \mathbb{E}\left\|\nabla_\omega F(\omega^{(t)}, \theta^*(\omega^{(t)}))\right\|_2^2.$$

Then we state the global convergence of the above mentioned optimization method.

**Theorem 3.** Suppose Assumptions [1], [3], [5], [6] hold. Given any $\epsilon > 0$, we take $\eta_\omega = \frac{\epsilon}{\sqrt{2}(2L_\omega + S_\omega^2/\mu)M_G}$, then we need at most

$$N = \widetilde{O}\left(\frac{(\rho_g^2/\mu + \lambda B_H)(L_\omega + S_\omega^2/\mu)M_G}{\epsilon^2}\right)$$

iterations to have $I_N < \epsilon$.

*Proof.* By employing the first inequality in Lemma [4], we have

$$F(\omega^{(t+1)}, \theta^*(\omega^{(t)})) - F(\omega^{(t)}, \theta^*(\omega^{(t)})) - \langle \nabla_\omega F(\omega^{(t)}, \theta^*(\omega^{(t)})), \omega^{(t+1)} - \omega^{(t)}\rangle$$

$$\leq \frac{L_\omega}{2}\left\|\omega^{(t+1)} - \omega^{(t)}\right\|_2^2. \tag{72}$$

Note that

$$\mathbb{E}\langle \nabla_\omega F(\omega^{(t)}, \theta^*(\omega^{(t)})), \omega^{(t+1)} - \omega^{(t)}\rangle$$

$$= \mathbb{E}\langle \nabla_\omega F(\omega^{(t)}, \theta^*(\omega^{(t)})), -\eta_\omega(\nabla_\omega F(\omega^{(t)}, \widetilde{\theta}^{(t)}) + \xi_\omega^{(t)})\rangle$$

$$= \mathbb{E}\langle \nabla_\omega F(\omega^{(t)}, \theta^*(\omega^{(t)})), -\eta_\omega(\nabla_\omega F(\omega^{(t)}, \theta^*(\omega^{(t)}) - \sigma_\theta^i) + \xi_\omega^{(t)})\rangle$$

$$\overset{(i)}{=} \mathbb{E}\langle \nabla_\omega F(\omega^{(t)}, \theta^*(\omega^{(t)})), -\eta_\omega(\nabla_\omega F(\omega^{(t)}, \theta^*(\omega^{(t)})) + \xi_\omega^{(t)})\rangle$$

$$\overset{(ii)}{=} \mathbb{E}\langle \nabla_\omega F(\omega^{(t)}, \theta^*(\omega^{(t)})), -\eta_\omega \nabla_\omega F(\omega^{(t)}, \theta^*(\omega^{(t)}))\rangle$$

$$= -\eta_\omega \mathbb{E}\left\|\nabla_\omega F(\omega^{(t)}, \theta^*(\omega^{(t)}))\right\|_2^2, \tag{73}$$

where $(i)$ comes from the unbiased property of $\widetilde{\theta}^{(t)}$ and the fact that $\nabla_\omega F(\omega, \theta)$ is linear in $\theta$, and $(ii)$ comes from the unbiased property of $\widetilde{f}_j(\omega, \widetilde{\theta})$. Now taking the expectation on both sides of (72) and plugging (73) in, we obtain

$$\mathbb{E}F(\omega^{(t+1)}, \theta^*(\omega^{(t)})) - F(\omega^{(t)}, \theta^*(\omega^{(t)})) + \eta_\omega \mathbb{E}\left\|\nabla_\omega F(\omega^{(t)},, \theta^*(\omega^{(t)}))\right\|_2^2 \leq \frac{L_\omega}{2}\eta_\omega^2 M_G. \tag{74}$$

Dividing both sides by $\eta_\omega$ and rearranging the terms in (74), we get

$$\mathbb{E}\left\|\nabla_\omega F(\omega^{(t)}, \theta^*(\omega^{(t)}))\right\|_2^2 \leq \frac{\mathbb{E}F(\omega^{(t)}, \theta^*(\omega^{(t)})) - F(\omega^{(t+1)}, \theta^*(\omega^{(t)}))}{\eta_\omega} + \frac{L_\omega}{2}\eta_\omega M_G$$

$$\leq \frac{\mathbb{E}F(\omega^{(t)}, \theta^*(\omega^{(t)})) - F(\omega^{(t+1)}, \theta^*(\omega^{(t+1)}))}{\eta_\omega} + \frac{L_\omega}{2}\eta_\omega M_G$$

$$+ \frac{\mathbb{E}F(\omega^{(t+1)}, \theta^*(\omega^{(t+1)})) - F(\omega^{(t+1)}, \theta^*(\omega^{(t)}))}{\eta_\omega}. \tag{75}$$

Now consider $F(\omega^{(t+1)}, \theta^*(\omega^{(t+1)})) - F(\omega^{(t+1)}, \theta^*(\omega^{(t)}))$, we have

$$F(\omega^{(t+1)}, \theta^*(\omega^{(t+1)})) - F(\omega^{(t+1)}, \theta^*(\omega^{(t)}))$$

$$= \langle G(\pi_{\omega^{(t+1)}}) - G(\pi^*), \theta^*(\omega^{(t+1)})\rangle - \langle G(\pi_{\omega^{(t+1)}}) - G(\pi^*), \theta^*(\omega^{(t)})\rangle$$

$$- \frac{\mu}{2}(\left\|\theta^*(\omega^{(t+1)})\right\|_2^2 - \left\|\theta^*(\omega^{(t)})\right\|_2^2)$$

$$= \langle \mu\theta^*(\omega^{(t+1)}), \theta^*(\omega^{(t+1)}) - \theta^*(\omega^{(t)})\rangle$$

$$- \frac{\mu}{2}\langle \theta^*(\omega^{(t+1)}) + \theta^*(\omega^{(t)}), \theta^*(\omega^{(t+1)}) - \theta^*(\omega^{(t)})\rangle$$

$$= \frac{\mu}{2}\left\|\theta^*(\omega^{(t+1)}) - \theta^*(\omega^{(t)})\right\|_2^2$$

$$= \frac{\mu}{2}\left\|\frac{1}{\mu}(G(\pi_{\omega^{(t+1)}}) - G(\pi_{\omega^{(t)}}))\right\|_2^2$$

$$\leq \frac{S_\omega^2}{2\mu}\left\|\omega^{(t+1)} - \omega^{(t)}\right\|_2^2. \tag{76}$$

Taking expectation on both sides of (76) with respect to the noise introduced by SGD, we have

$$\mathbb{E}F(\omega^{(t+1)}, \theta^*(\omega^{(t+1)})) - F(\omega^{(t+1)}, \theta^*(\omega^{(t)})) \le \frac{S_\omega^2}{2\mu}\eta_\omega^2 M_G.$$

Summing the equation(75) up, we have

$$\sum_{t=1}^N \mathbb{E}\left\|\nabla_\omega F(\omega^{(t)}, \theta^*(\omega^{(t)}))\right\|_2^2$$

$$\le \frac{1}{\eta_\omega}\sum_{i=1}^N \mathbb{E}F(\omega^{(t)}, \theta^*(\omega^{(t)})) - F(\omega^{(t+1)}, \theta^*(\omega^{(t)})) + \frac{L_\omega}{2}N\eta_\omega M_G$$

$$\le \frac{1}{\eta_\omega}\sum_{i=1}^N \mathbb{E}F(\omega^{(t)}, \theta^*(\omega^{(t)})) - F(\omega^{(t+1)}, \theta^*(\omega^{(t+1)}))$$

$$+ \frac{1}{\eta_\omega}\sum_{i=1}^N \mathbb{E}F(\omega^{(t+1)}, \theta^*(\omega^{(t+1)})) - F(\omega^{(t+1)}, \theta^*(\omega^{(t)})) + \frac{L_\omega}{2}N\eta_\omega M_G$$

$$\le \frac{1}{\eta_\omega}\sum_{i=1}^N \mathbb{E}F(\omega^{(t)}, \theta^*(\omega^{(t)})) - F(\omega^{(t+1)}, \theta^*(\omega^{(t+1)}))$$

$$+ \frac{1}{\eta_\omega}\sum_{i=1}^N \frac{S_\omega^2}{2\mu}\eta_\omega^2 M_G + \frac{L_\omega}{2}N\eta_\omega M_G.$$

Dividing both sides of the above equation by $N$, we get

$$\min_{1\le t\le N} \mathbb{E}\left\|\nabla_\omega F(\omega^{(t)}, \theta^*(\omega^{(t)}))\right\|_2^2 \le \frac{|F(\omega^{(1)}, \theta^*(\omega^{(1)})) - \mathbb{E}F(\omega^{(N+1)}, \theta^*(\omega^{(N+1)}))|}{N\eta_\omega}$$

$$+ (\frac{L_\omega}{2} + \frac{S_\omega^2}{2\mu})\eta_\omega M_G.$$

Since $|F(\omega^{(1)}, \theta^*(\omega^{(1)})) - \mathbb{E}F(\omega^{(N+1)}, \theta^*(\omega^{(N+1)}))| \le 2B_F$. Take $\eta_\omega = \sqrt{\frac{2B_F}{(L_\omega/2 + S_\omega^2/2)M_G N}}$, then we have

$$\min_{1\le t\le N} \mathbb{E}\left\|\nabla_\omega F(\omega^{(t)}, \theta^*(\omega^{(t)}))\right\|_2^2 \le 4\sqrt{\frac{B_F(L_\omega + S_\omega^2/\mu)M_G}{N}},$$

where $B_F = \frac{12\rho_g^2}{\mu} + \lambda M_H$. This implies that when $\eta_\omega = \frac{\epsilon}{\sqrt{2}(2L_\omega + S_\omega^2/\mu)M_G}$, we need at most

$$N = \widetilde{O}\left(\frac{(\rho_g^2/\mu + \lambda M_H)(L_\omega + S_\omega^2/\mu)M_G}{\epsilon^2}\right)$$

such that $I_N < \epsilon$. $\hfill\square$

