# OpenReview forum: "On Computation and Generalization of Generative Adversarial Imitation Learning"
_ICLR.cc/2020/Conference — Accept (Poster)_

### Official Review · AnonReviewer2 · 2019-10-23
**Official Blind Review #2**

**Rating:** 6

**Review:**

Summary: This paper presents theoretical analysis of the generalization properties of GAIL, as well as the local convergence of the traditional minibatch SGD applied to its min-max optimization problem (without assuming convex-concave structure of the game).  Specifically, the authors first prove a generalization bound for GAIL that characterizes how well optimizing the empirical GAIL objective minimizes the true population version (in terms of an "R-distance" they use the characterize the expected distributions induced by the underlying policies for a given reward class).  Second, they prove a convergence result for minibatch SGD applied to the min-max game, showing that the method will converge to a stationary point regardless of any convex-concave structure.

Comments: Before I begin, I should add that several elements of the paper were a bit difficult for me to follow, so I'm happy for the authors to correct any factual errors that I might make.  Overall, I think that this is an interesting, if somewhat incremental and technical paper about the generalization and convergence of GAIL.

First, on the generalization aspects, the methodology here seems to largely parallel Arora et al.'s analysis of generalization in GANs.  The main technical steps seem to be, 1) some effort in determining the proper distance to use in the first place, and how to define generalization, which they do via the R-distance, and then 2) on the more technical side, overcoming the fact that trajectories do not provide iid samples as is the case in the GAN analysis.  This later difficulty is overcome via the independent block technique, which allows one to bound the relevant population quantities of interest by sampling from subsampled blocks of the original trajectory.

Second, on the convergence side, the main result here seems to be a generic convergence result for minibatch SGD applied to a non-convex-concave game.  I may be missing something here, but it doesn't seem like there is any real specificity to the GAIL objective, but rather this would apply to any such min-max problem (I suppose also in the case where there is an L2 projection in the gradient step); while there is some discussion of the chain mixing properties, this seems largely needed just to provide bounds on certain constants.  The proof is rather technical (I admit I didn't go through it in much detail) and even the sketch doesn't provide a great intuition about what might make this problem harder than proving convergence of traditional SGD in the non-convex pure minimization case.  And for example, even convex-concave games have well-known pathologies when running gradient descent, such as cycling around an optimal point, and I didn't understand whether anything was being done to explicitly account for this, or if one of their assumptions essentially just avoids this possibility (maybe the mixing properties prevent this?).

On the whole, despite what seems to me to be a somewhat incremental nature, I'm still leaning toward accepting the paper: technical analysis like this is good to have, and the setting is distinct enough from past e.g., GAN work, that I believe it stands on its own.  I do think the clarity of the paper can be improved, as well.  This includes some simple elements like spacing out the equations better in Section 4, which were currently very condensed and difficult to read (and there is plenty of space to make the paper longer).  But I also would have liked a bit higher-level explanation of the challenges involved in proving convergence (like avoiding cycling), rather than just highlighting the functional elements of the proof.  I actually think the generalization section did this relatively well, in regards to the setting up their choice of IPM in contrast to the GAN work, but it was lacking in the convergence section.

**Experience Assessment:**

I have published one or two papers in this area.

**Review Assessment: Checking Correctness Of Derivations And Theory:**

I assessed the sensibility of the derivations and theory.

**Review Assessment: Checking Correctness Of Experiments:**

I assessed the sensibility of the experiments.

**Review Assessment: Thoroughness In Paper Reading:**

I read the paper thoroughly.

---

> ### Author Response · Authors · 2019-11-15
> **To Reviewer #2**
>
> We thank you for your valuable comments and questions.
>
> Q: Regarding convergence of mini-batch SGD.
>
> A: We consider alternating stochastic gradient descent ascent algorithm for solving the minimax optimization of GAIL. This is quite different from SGD for nonconvex pure minimization. Specifically, the inner maximization problem induces a function of policy denoted by $G(\omega) = \max_{\|\theta\|_2\leq\kappa}\mathbb{E}_{\tilde{\pi}_{\omega}}[\tilde{r}_{\theta}(s,a)]-\mathbb{E}_{\pi^{*}}\tilde{r}_{\theta}[(\psi_s,\psi_a)] - \lambda H(\tilde{\pi}_{\omega})- \frac{\mu}{2}\|\theta\|_2^2$. When alternatively updating the policy parameter $w$, the gradient $\nabla_{\omega}\tilde{f}(\omega^{(t)},\theta^{(t+1)})$ is not the true gradient $\nabla G(\omega)$, since the reward parameter $\theta$ at each iteration is not optimal with respect to the policy. Even worse, the stochastic gradient $\nabla_{\omega}\tilde{f}$ can be misleading, when the inner product of $\nabla_{\omega}\tilde{f}$ and $\nabla_{\omega} G$ is negative. This causes the minimax optimization unstable and it is therefore, technically more involved to analyze the convergence of minimax optimization.
>
> We construct a potential function to prove the convergence of alternating stochastic gradient descent ascent. We show the potential function monotonically decreasing along the solution path in Lemma 1. This relies on the following important regularity conditions (Assumptions 4 and 5) in our problem: 1) The inner maximization problem is strongly concave due to the quadratic regularizer; 2) The mixing condition in Assumption 5 ensures the objective function is Lipschitz and smooth (Lipschitz gradient) with respect to the policy as well as the reward; 3) The step sizes are properly chosen as in Theorem 2. These benign conditions are often missing in many convex-concave games, in which existing literature has shown that gradient descent ascent fails to converge (e.g., bilinear setting).
>
> The proof sketch of Theorem 2 has been revised in the latest version.

---

### Official Review · AnonReviewer1 · 2019-10-23
**Official Blind Review #1**

**Rating:** 6

**Review:**

The submission provides theoretical analysis of GAIL regarding its generalization and convergence properties.
The first part establishes a probabilistic upper bound on the change of the worst-case regret (over the possible reward function) when shifting from the empirical expectation of the expect reward to the true expectation.
The second part considers the convergence properties of GAIL when alternating mini-batch stochastic gradient updates on the policy and discriminator. This section shows that number of iterations required to achieve a "sub-stationarity" J < epsilon is in O(1/epsilon). The proof assumes (vanilla) policy gradient updates, and seems to be further restricted to linear discriminators with bounded weights.
Experiments on Acrobot, Hopper and MountainCar compare learning curves for a 3-layer neural network reward function, and linear reward functions. The linear reward function uses the neural network architecture but does not optimize the first two layers (and is thus linear in random features). Furthermore--for the linear reward function--single gradient updates are compared with 10 gradient updates per iteration. The experiments indicate that single updates perform similar to 10 updates and that the linear reward function converges more stable than the neural network. The neural network, however, can achieve slightly better performance on the considered problems. These results are consistent with the provided theory.

Contribution/Significance:
I think that the theoretical properties of GAIL and related adversarial IL and IRL methods are not yet sufficiently understood. Both achieving stable convergence and generalization from limited number of trajectories can be difficult in practice, so there is high interest in theoretic analysis of these methods. I am not aware of similar analysis of GAIL.

Soundness:
I did not have the time to fully verify the proofs, so I only skimmed the appendix and focused on the proof sketches in the actual manuscript. The assumptions seem reasonable and I could not find errors in the proof sketches. I am having some problems with the proof sketch of Theorem 1. I am not sure about the meaning of $\mathbb{E} \phi(s')$ at the last line of page 4, which is unfortunately not rigorously defined. I assume the expectation is with respect to the sampled blocks. So wouldn't this term be a functional of r? However, I assume the supremum is only w.r.t. the first term (what would be a supremum over an inequality anyway?). Also it seems like the term could be subtracted from both sides of the inequality. Some hints would be highly appreciated here.

On a minor note, I think that min and max should be swapped in Equation 1 and 2 (r corresponding to a reward, not cost).

Presentation/Clarity:
I don't have a strong mathematical background and would not say that the paper is fully clear to me. However, this is rather caused by the nature of the paper. Indeed, I think that the paper is well written and relatively clear.

Evaluation:
The paper only contains a very short experiment section, however, this is reasonable given that contributions are theoretical. I would be interested in some insights on the strong oscillatory behavior of the NN-reward on the acrobot task. I noticed such behavior with GAIL also on more complex tasks and sometimes it would not even recover as quickly as in the acrobot plot. Is this caused by overfitting of the discriminator?

Assessment:
I think that the paper could be an interesting contribution for ICLR. However, my confidence is rather small here.

**Experience Assessment:**

I have read many papers in this area.

**Review Assessment: Checking Correctness Of Derivations And Theory:**

I assessed the sensibility of the derivations and theory.

**Review Assessment: Checking Correctness Of Experiments:**

I assessed the sensibility of the experiments.

**Review Assessment: Thoroughness In Paper Reading:**

I read the paper at least twice and used my best judgement in assessing the paper.

---

> ### Author Response · Authors · 2019-11-15
> **To Reviewer #1**
>
> We appreciate your valuable comments and questions.
>
> Q: Regarding the proof sketch of Theorem 1.
>
> A: The expectation on $\phi$ is indeed taken with respect to the sampled blocks.
>
> We have elaborated on the proof sketch to clarify the last inequality on page 4 and correct a typo. In more detail, we show $\mathbb{P}(\sup_r \phi_1 - \mathbb{E}[\sup_r \tilde{\phi}_1] \geq \epsilon - \mathbb{E}[\sup_r \tilde{\phi}_1]) \leq \mathbb{P}(\sup_r \tilde{\phi}_1 - \mathbb{E}[\sup_r \tilde{\phi}_1] \geq \epsilon - \mathbb{E}[\sup_r \tilde{\phi}_1]) + C \beta T / (\log T + \log (1/\delta))^{1/\alpha}$. Here we augment the inequality with $\mathbb{E}[\sup_r \tilde{\phi}_1]$ term, since the different $\sup_r \tilde{\phi}_1 - \mathbb{E}[\sup_r \tilde{\phi}_1]$ can be controlled using the empirical process technique for independent random variables ($\tilde{\phi}_1$ takes samples from independent blocks).
>
> Q: I think that min and max should be swapped in Equation 1 and 2 (r corresponding to a reward, not cost).
>
> A: The min-max formula in equations 1 and 2 are in the correct order. This is identical to the formula derived in equations (14) and (15) in [1]. Here the inner maximization measures the discrepancy between the expert and learned policies with respect to all possible reward functions, which yields the so-called R-distance in Definition 2. Then the outer minimization optimizes the learned policy to mimic that of the expert under the R-distance.
>
> If $r$ is a cost function, GAIL still takes the min-max formula. In this case, the inner maximization aims to find the max difference between expert and learned policies with respect to all possible costs. Note that GAIL is different from inverse reinforcement learning (IRL): GAIL attempts to recovery the expert policy with respect to all possible rewards (costs), while IRL fits a cost function using the given expert policy, and the formula is max-min as given in equation (1) in [1].
>
> Q: Strong oscillatory behavior of NN-reward.
>
> A: We believe that the strong oscillatory is due to the minimax optimization of training GAIL. Such a stability issue is also widely observed in GAN’s training [2, 3]. From a theoretical perspective, the inner maximization problem induces a function of policy denoted by $G(\omega) = \max_{\|\theta\|_2\leq\kappa}\mathbb{E}_{\tilde{\pi}_{\omega}}[\tilde{r}_{\theta}(s,a)]-\mathbb{E}_{\pi^{*}}\tilde{r}_{\theta}[(\psi_s,\psi_a)] - \lambda H(\tilde{\pi}_{\omega})- \frac{\mu}{2}\|\theta\|_2^2$. When alternatively updating the policy parameter $w$, the gradient $\nabla_{\omega}\tilde{f}(\omega^{(t)},\theta^{(t+1)})$ is not the true gradient $\nabla G(\omega)$, since the reward parameter $\theta$ at each iteration is not optimal with respect to the policy. Since NN-reward does not induce strong concavity in $G$, the stochastic gradient $\nabla_{\omega}\tilde{f}$ can be misleading, when the inner product of $\nabla_{\omega} f$ and $\nabla_{\omega} G$ is negative. Therefore, we expect to see oscillatory during the training. We currently do not have a concrete explanation on why the oscillatory recovers in the acrobot plot. This behavior is task dependent (different tasks induce different objectives with varying curvature) and we leave it for future investigation.
>
> Reference
> [1] Ho, Jonathan, and Stefano Ermon. "Generative adversarial imitation learning." In Advances in neural information processing systems, pp. 4565-4573. 2016.
> [2] Mescheder, Lars, Andreas Geiger, and Sebastian Nowozin. "Which training methods for GANs do actually converge?." arXiv preprint arXiv:1801.04406 (2018).
> [3] Miyato, Takeru, Toshiki Kataoka, Masanori Koyama, and Yuichi Yoshida. "Spectral normalization for generative adversarial networks." arXiv preprint arXiv:1802.05957 (2018).

---

> > ### Comment · AnonReviewer1 · 2019-11-15
> > **Thanks for clarifications**
> >
> > Thank you for the clarifications. I did not have time yet to look at them in detail, but they will likely be useful during the post-rebuttal.

---

### Official Review · AnonReviewer3 · 2019-10-25
**Official Blind Review #3**

**Rating:** 6

**Review:**

This paper investigates the theoretical support for Generative Adversarial Imitation Learning (GAIL).   Specifically, two main points are shown: (1) For general reward parameterization, the generalization of GAIL can be guaranteed, as long as the class of the reward functions is properly controlled; (2) When the reward is parameterized as a reproducing kernel function, GAIL can be efficiently solved by stochastic first order optimization algorithms.

Numerical experiments are provided on three classic continuous control tasks. RL algorithms are generally questioned in terms of reproducibility. Does the variance of different runs have an impact on the validation of the proposed theory?

**Experience Assessment:**

I do not know much about this area.

**Review Assessment: Checking Correctness Of Derivations And Theory:**

I did not assess the derivations or theory.

**Review Assessment: Checking Correctness Of Experiments:**

I assessed the sensibility of the experiments.

**Review Assessment: Thoroughness In Paper Reading:**

I made a quick assessment of this paper.

---

> ### Author Response · Authors · 2019-11-15
> **To Reviewer #3**
>
> We thank you for your valuable comments and questions.
>
> Q: Does the variance of different runs have an impact on the validation of the proposed theory?
>
> A: The variance of different runs comes from the stochastic gradient in each iteration (equations (3) and (4)). Our proposed theory already takes the variance into account. Specifically, under the variance bounded assumption (Assumption 4), we show that the potential function is monotonically decreasing (Lemma 1). Based on this, we prove the convergence of our alternating stochastic gradient descent ascent algorithm.
>
> Moreover, our experiment validates our proposed computational theory. The plotted curves in Figure 1 are average reward obtained by multiple independent evaluations of the learned policy in the environment. We see that the plotted curves are well concentrated around its average performance, despite the variation in each trajectory. After sufficiently many iterations, the average reward converges, which corroborates Theorem 2.

---

### Decision · Program_Chairs · 2019-12-19

**Decision:**

Accept (Poster)

**Comment:**

The paper provides a theoretical analysis of the recent and popular Generative Adversarial Imitation Learning (GAIL) approach. Valuable new insights on generalization and convergence are developed, and put GAIL on a stronger theoretical foundation. Reviewer questions and suggestions were largely addressed during the rebuttal.